# Updated Applications of Stem Cells in Hypoplastic Left Heart Syndrome

**DOI:** 10.3390/cells14171396

**Published:** 2025-09-06

**Authors:** Rui Xiao, Haleema Darr, Zarif Khan, Qingzhong Xiao

**Affiliations:** 1Norfolk and Norwich University Hospital NHS Foundation Trust, Colney Ln, Norwich NR4 7UY, UK; rui.xiao@esneft.nhs.uk; 2University Hospital Coventry and Warwickshire, Clifford Bridge Rd, Coventry CV2 2DX, UK; haleema.darr@uhcw.nhs.uk; 3University College London Hospital NHS Foundation Trust, 250 Euston Road, London NW1 2PG, UK; zarif.khan@esneft.nhs.uk; 4William Harvey Research Institute, Faculty of Medicine and Dentistry, Queen Mary University of London, London EC1M 6BQ, UK

**Keywords:** hypoplastic left heart syndrome, congenital heart disease, stem cell, pluripotent stem cell, induced pluripotent stem cell, cardiac organoid, engineered cardiac tissue, stem cell therapy, drug screening, 3D/4D bioprinting

## Abstract

Hypoplastic left heart syndrome (HLHS) is a severe congenital heart disease affecting 2–3 neonates every 10,000 live births. While prior research has highlighted associations of HLHS with specific chromosomal abnormalities and genetic mutations, the precise pathophysiology remains elusive. Despite early surgical intervention potentially allowing most HLHS patients to survive their critical heart disease with a single-ventricle physiology, patients frequently experience complications of arrhythmias and right ventricular heart failure, culminating in the need for an eventual heart transplant. Scarcity of suitable donors combined with limited understanding of mechanisms of development highlights the need for furthering our understanding of HLHS and alternative treatment options. Over the past decades, stem cell research has significantly advanced our understanding of cardiac conditions, repair, development, and therapy, opening the door for a new exciting field of regenerative medicine in cardiology with significant implications for HLHS. This review serves to provide a comprehensive overview of a much focused-on area related to HLHS. Specifically, we will first discuss the key pathophysiological basis and signalling molecules of HLHS. We then outline the emerging role of stem cell-based therapy, with a focus on adult stem cells and pluripotent stem cells (PSCs) in uncovering the pathophysiology of HLHS and optimising future treatment directions. Finally, we will also explore the latest and possible future directions of stem cell-derived techniques such as cardiac organoids and bioengineering cardiac tissues and their utility for investigating disease mechanisms, drug screening, and novel therapy for HLHF.

## 1. Introduction

Hypoplastic left heart syndrome (HLHS) is a severe and complex congenital heart disease (CHD) that affects 1 in 3995 live births in the USA and constitutes roughly 2–3% of all CHDs [1]. The disease is characterised by the underdevelopment of left-sided cardiac structures including left ventricle, mitral valve, aortic valve, and ascending aorta [2]. The resulting anatomical deficiency prevents effective systemic circulation, thus rendering patients with a heart that is unable to deliver oxygenated blood throughout the body, often leading to mortality without early surgical intervention [3].

The current standard of care for HLHS patients involves a series of staged surgical procedures aimed to overcome left-sided cardiac deficiencies [2,4,5] (Figure 1). The initial Norwood procedure, usually performed in the first week of life, reconstructs the underdeveloped aorta with the pulmonary artery to form a neoaorta. This allows the right ventricle (RV) to supply the systemic circulation [6]. At the same time, surgeons will insert a shunt from the RV to supply the pulmonary circulation and create an artificial septum between the right and left atrium to allow mixing of oxygenated and deoxygenated blood [7]. Subsequently, the bidirectional Glenn procedure, normally occurring at 4–6 months, will connect the superior vena cava to the pulmonary arteries, thereby bypassing the heart and resulting in a reduced RV workload [8]. Similarly, the final palliative stage termed the Fontan procedure, performed between 18 months and 3 years of age, reconnects the inferior vena cava to the pulmonary artery, allowing all deoxygenated blood to enter the lungs directly [9]. Overall these surgical palliation procedures allow HLHS patients to achieve an RV that is capable of supplying fully oxygenated blood to the systemic circulation, thereby extending their life expectancy and improving their quality of life [10], although it is still far off from that of normal individuals [11]. Despite surgical advances enabling survival of an otherwise fatal condition, long-term complications such as arrhythmias [12,13] and eventual RV failure occur [14] in a significant proportion of patients. Many HLHS patients even with successful staged palliation surgeries will eventually require heart transplantation [15], which is significantly limited by the availability of suitable donors and comes with significant complications and risks of its own. This coupled with the limited regenerative capacity of the myocardium highlights the need for alternative therapeutic options.

The last two decades have seen huge strides towards addressing these limitations with the emergence of regenerative therapies using stem cell techniques. Multiple sources of viable adult stem cells have been discovered within bone marrow, adipose tissue, and umbilical cord blood [16,17,18]. These stem cells have shown translational potential within a wide range of pathologies including HLHS, with multiple published and ongoing preclinical and clinical trials investigating their potential usage in HLHS [19]. Early clinical results have demonstrated the feasibility and safety of stem cell application in HLHS patients such as in the ELPIS trials [20], paving the way for later phased clinical trials to investigate the efficacy of stem cell therapy.

Other than adult stem cells, pluripotent stem cells (PSCs) including induced PSCs (iPSCs) and embryonic stem cells (ESCs) offer an excellent alternative possibility. PSC-derived cardiomyocytes (CMs) [21] have been used extensively to study disease modelling, pathogenesis, and potential therapeutic applications [22,23,24,25]. More recently, the discovery of cardiac organoids and improvements in cardiac tissue engineering with the introduction of 3D bioprinting has seen a flux of studies incorporating these elements into the study of HLHS [26,27]. In this review, we aim to explore how recent stem cell technologies have redefined our understanding of HLHS and potentially provide innovative approaches to the treatment of HLHS beyond surgery.

## 2. Key Pathophysiological Basis and Signalling Molecules of HLHS

As nicely discussed by Li et al. [28], cardiac development is a complex and intricate process involving the formation of a primitive heart tube, followed by looping, septation, and the development of chambers and valves. In short, it initially begins between day 15–17 of embryogenesis with the formation of cardiac mesoderm in the primitive streak that migrate and differentiate into the heart’s components to form the two heart-forming regions (HFRs), each consisting of the first heart field (FHF) on the anterior lateral side and the secondary heart field (SHF) on the anterior medial side [29,30]. These two HFRs then merge to form the cardiac crescent and eventually fold to form the primitive heart tube. The FHF contributes to the formation of the left ventricle (LV) and parts of the atria and right ventricle (RV) whilst the SHF contributes to the development of the outflow tract, the majority of the RV, and the atria/venous pole. The primitive heart tube undergoes looping which transforms the structure from that of a linear tube into a structure with distinct chambers. Subsequently, neural crest cells migrate into the cushions to form the aorticopulmonary septum [31] whilst endocardial cushions give rise to the heart valves and septa [32].

As discussed in an elegant review [27], emerging evidence generated from different animal models such as those in chick, mouse, lamb, and pig, as well as several in vitro models using iPSCs has uncovered some important insights into hemodynamic and genetic factors influencing HLHS. Hemodynamic forces and blood flow mechanics are fundamental to cardiac development [33]. Accordingly, a popular concept termed as the “no flow, no grow” hypothesis has been widely appreciated in the HLHS field. It suggests that abnormal cardiac valves could limit blood flow, leading to LV underdevelopment. However, the cause of the initial insult(s) that leads to perturbation in blood flow remains unknown. It has been suggested that any initial genetic perturbations in the left heart valvular or ventricular primordial tissue or environmental disturbance in utero could impair cardiac cell numbers, shape, and contractility, altering hemodynamic mechanical forces and mechanosensitive signalling pathways in the developing heart. Such alterations may further impair cardiac cell number and functional properties, ultimately producing the clinical HLHS phenotype [34]. Indeed, a multitude of potential developmental and signalling pathways alongside growth and transcription factors have been proposed as contributory or even pathogenic components to HLHS [35] (Table 1). A foundational understanding of the finely balanced genetic and developmental signalling pathways is therefore the key to interpreting recent insights into HLHS pathogenesis.

### 2.1. Key Signalling Pathways for Heart Development and HLHS

The complex process of cardiogenesis relies on finely tuned spatial and temporal development of the heart [103]. This is achieved via interactions between numerous transcription factors and signalling molecules as detailed extensively in Li et al.’s review on molecular mechanisms on cardiac development [28]. Here, we will summarise the most important molecular pathways and transcription factors with additional focus on HLHS relevant targets (Table 1).

One of the most extensively studied pathways for cardiac development and HLHS is the WNT/β-catenin signalling cascade [104]. Activation of the canonical WNT pathway leads to an accumulation of β-catenin via frizzled receptors [105,106]. This induces the development of mesoderm from the primitive streak which will subsequently form the FHF and the SHF. Wnt/β-catenin signalling plays an important role in the regional expansion of both FHF and SHF-derived ventricular cardiomyocytes [107], inferring a critical role of this signal pathway in cardiac development which has been clearly demonstrated in multiple genetic mutation models [108,109,110,111]. Specifically, loss of Wnt2 results in a reduction in posterior SHF progenitors and defects in the cardiac inflow tract [110]. Wnt11 mutants show cardiac OFT defects [111], as well as reduced areas of the compact and trabeculated myocardium with hypoplasic and thinner LV and RV [109], resembling the developmental defects in HLHS. Additionally, Wnt5a-deficient mice exhibit failure of cardiac progenitor cells to extend into the arterial and venous poles, leading to both outflow tract and atrial septation defects [112], another signature of developmental defects in HLHS. These findings indicate that WNT signalling plays a crucial yet complex role in early cardiac development and HLHS pathogenesis; however, its exact contribution to HLHS pathogenesis remains to be fully elucidated.

Another key signalling pathway implicated in the development of the heart is retinoic acid (RA). Under normal circumstances, RA produced by the retinaldehyde dehydrogenase enzymes (RALDH) acts as a morphogen which governs the development of tissue patterning and cardiac defects [113]. By exerting its effects on retinoic acid receptors (RARs and RXRs), RA establishes correct anterior–posterior (AP) axis patterns in cardiac mesoderm, which defines atrial and ventricular regions [114,115]. RA signalling plays essential roles in cardiac development, from the early formation of AP boundaries of the cardiac mesoderm to the development of the epicardium and the subsequent formation of normal cardiac morphology [28]. Particularly, RXRα knockout embryos display complex defects, including ventricular septal, atrioventricular cushion, and conotruncal ridge defects, with a double outlet right ventricle, aorticopulmonary window, and persistent truncus arteriosus [116]. Deficiency of Raldh2, a protein controlling the primary source of RA during embryonic development, results in incorrect heart looping and severe impairment of atrial and sinus venosus development [50]. Interestingly, RA signalling is required to restrict the size of the FHF within the anterior lateral plate mesoderm, and its disruption results in increased FHF-derived CMs and a progressive decrease in SHF progenitors at the arterial pole as the heart tube elongates, leading to small OFTs at later stages [117]. Additionally, RA has been proven to prevent overgrowth of cardiac progenitor regions and regulate epithelial-to-mesenchymal transition [118]. Finally, RA has been documented to interact with several key factors such as bone morphogenetic proteins (BMPs) and NKX2.5 implicated in HLHS pathogenesis [113], implying a potential role in HLHS aetiology. Although findings from the abovementioned studies clearly confirm a critical role for RA signalling in early cardiac development, direct evidence to support a functional role for the RA signalling pathway in HLHS pathogenesis is still missing.

Amongst the key cardiac developmental pathways, disruption in NOTCH signalling remains one of the most heavily researched pathways with strong evidence supporting its role in HLHS pathogenesis. The normal function of the NOTCH pathway begins with Jagged and delta-like ligands binding to NOTCH 1-4 to facilitate activation of downstream genes via the NOTCH intracellular domain (NICD) [36]. During cardiac development, activation of NOTCH signalling initially prevents expression of myocardial genes and inappropriate early cardiac differentiation [119], thereby keeping mesoderm in its progenitor state. Subsequently, NOTCH signalling interacts with other pathways such as WNT and BMP, promoting SHF differentiation into outflow tracts (OFTs) and right heart structures. In addition, NOTCH has been described in detail by many studies on its role in ventricular trabeculation, normal endocardial cushion placement, valve thickness, and proper ventricular growth [120,121]. Gene mutations in the NOTCH pathway have been associated with a variety of CHDs including bicuspid aortic valve stenosis, VSD, overriding aorta, hypoplastic left heart, and incomplete right ventricular development [28,122]. Mouse embryos with epicardium-specific NOTCH1 ablation exhibit disruption of coronary artery differentiation, thinner myocardium wall, and decreased CM proliferation. Interestingly, ectopic NOTCH1 activation also disrupts epicardium development and causes thinning of ventricular walls [123], indicating that a fine balance in NOTCH1 signalling is crucial for myocardium growth. Despite the fact that close associations between genetic mutations and HLHS were widely reported, the exact and precise contributions of NOTCH signalling to developmental cardiac defects observed in HLHS remain unknown.

BMP is a family of growth factors that forms a subset of the much larger transforming growth factor beta (TGFβ) family. They act as signalling proteins that interact with many of the pathways mentioned previously to ensure normal cardiac development. Different members of the BMP family are instrumental in different aspects of cardiac development. BMP2 and BMP4 working in tandem with WNT, FGF, and other growth factors help to guide mesodermal cells to a cardiac fate [41]. These BMPs are required for cardiac mesodermal development and proliferation of endocardial cushions which eventually form the heart valves, structures that are underdeveloped in HLHS. BmpR1a deficiency in early mesoderm progenitors that contribute to both FHF and SHF results in absence of the entire cardiac crescent and later the primitive ventricle and abolishes FHF marker genes’ expression (eHand and Tbx-5) [124], inferring a critical role for BmpR1a in FHF cell specification and ventricle development. Genetic deletion of BMP10 confirmed an important role for BMP10 in regulating cardiac growth and chamber maturation, as evidenced by reduced CM proliferation and thinner ventricular walls in BMP10-null mice [125]. Moreover, studies also demonstrate that BMP10 is critical in trabeculation [126] and CM proliferation [125] with its absence in mice leading to ventricular wall underdevelopment and increased lethality rates during embryogenesis [45]. These findings highlight the multifaceted roles of numerous members of the BMP family in cardiogenesis and serve as a potential source of pathogenesis for HLHS studies. However, such a notion needs further genetic validation.

While this section does not aim to provide a comprehensive review on the topic of cardiac signalling pathways, the selected pathways offer helpful context for subsequent discussion on the use of stem cell technologies in uncovering HLHS mechanisms. Table 1 listed some key cardiac developmental signalling pathways discovered thus far, including some yet to be associated with HLHS. Beyond signalling pathways, some transcription factors have also been investigated in the pathogenesis of HLHS with many of them thought to interact closely with cardiac development pathways.

### 2.2. Transcription Factors Underpinning Cardiac Development and HLHS

Part of the reason why HLHS aetiology remains difficult to elucidate is the large quantity of transcription factors, growth factors, and signalling pathways involved in normal cardiac development (Table 1). They work synergistically with multiple molecules that must function in tandem [127,128,129]. A single factor can exert multiple permissive and inhibitory effects on other pathways in a precise time-dependent manner with dysregulation at any stage leading to downstream effects [127,130,131]. Key transcription factors implicated in normal cardiac function include HAND1/2 [65], NKX2.5 [69], GATA4 [132], MYRF [74], ISL1 [133], TBX1/5 [134], and MEF2C [135], amongst many others. In this section, we will break down the role of transcription factors with the most extensive literature surrounding their involvement in HLHS. The remaining factors as listed in Table 1 are still crucial for cardiogenesis, although their role in HLHS remains less clear.

NKX2.5 is an important homeobox transcription factor expressed in early cardiac progenitors and is responsible for heart tube formation, chamber specification, and heart looping [136]. Evidence demonstrates that interactions between NKX2.5 and other transcription factors promote CM differentiation [137], with NKX2.5 and HAND2 interactions crucial for ventricular identity [65]. NKX2.5 knockout mice [70] and zebrafish [138] exhibited severe cardiac defects including abnormal conduction system, impaired heart tube, and ventricle development [139]. In patient cohort studies, NKX2.5 defects have been associated with various CHDs [140]; however, no specific strong association with HLHS was demonstrated [72]. This has led to the use of patient-derived iPSC being used as an alternative tool to investigate the exact role NKX2.5 has in HLHS [141].

Similarly, HAND1, a basic helix–loop–helix (bHLH) transcription factor, works within a complex network of transcription factors such as NKX2.5 and GATA4 [142] to regulate CM differentiation [143]. Crucially for HLHS, HAND1 plays a critical role in left heart development by specifying left ventricle identity [144]. Genetic disruptions of HAND1 in animal models confirmed its critical role in cardiac development, as evidenced by the removal of HAND1 leading to severe cardiac malformations including poorly formed or missing left ventricles [144,145], resembling features seen in HLHS. In human studies, reports using patient cardiac tissue sequencing identified a frequent frameshift mutation (A126fs) leading to a truncated HAND1 protein [67]. However, follow-up studies using mouse models engineered with the same mutations were unable to replicate a hypoplastic left ventricle [68]. Instead, mice with the mutation demonstrated increased embryonic lethality and structural abnormalities. These results reinforce the importance of HAND1 in normal cardiac development whereas its specific role in HLHS may be more contributory rather than a monogenic cause which is consistent with current understanding of HLHS aetiology.

Beyond transcription factors, some vital structural and regulatory proteins such as MYH6, LRP2, and GJA1, amongst many others, have also been proposed as alternative contributory factors for HLHS [35]. In addition, multiple chromosomal syndromes such as Turner’s, Jacobsen’s, DiGeorge, Holt–Oram, and Edward’s syndromes have been associated with HLHS to various degrees [2,146], further highlighting the multifactorial aetiology of HLHS and why it has been so difficult to unravel.

Finally, it worth mentioning that evidence from genetic screening showed that HLHS is a heterogeneous genetic condition with complicated congenital heart defects [147] and is characterised by the underdevelopment of the LV, mitral and aortic valve, and ascending aorta. The nature of the genetic heterogeneity of HLHS may underpin the clinically observed differences in the severity of left-sided cardiac structures manifesting in HLHS patients. The paucity of knowledge about the genetic causes of HLHS and the lack of a genetic animal model that can faithfully recapitulate all the clinical phenotypes of HLHS patients may be partially attributed to our current poor understanding of the aetiology of HLHS, making it extremely difficult to identify effective treatment for HLHS. Using mouse forward genetics, Lo et al. [147,148,149] were the first to isolate HLHS mutant mice and identify HLHS-causing genes. They found that mutations from seven HLHS mouse lines showed multigenic enrichment in ten human chromosome regions linked to HLHS and confirmed the digenic causes of HLHS [147]. Specifically, they reported that mutation in *Sap130* drives LV hypoplasia, while mutation in *Pcdha9* causes valvular defects associated with HLHS, both signature HLHS defects. Accordingly, they proposed a new paradigm in which HLHS may arise in a modular fashion, mediated by multiple genetic mutations [147,148,149]. Despite these new advancement in genetic studies, whether defects in the development myocardium, endocardium, valves alone, or all of them are the primary developmental cause (s) for HLHS remains to be elucidated. Further investigations into these aspects will be crucial when considering how the use of stem cells might help our understanding of the pathological causes of HLHS.

## 3. Application of Stem Cell in HLHS as an Adjunct Treatment

Unsurprisingly, no effective medication is currently available for HLHS patients due to its complicated and multifactorial aetiology. Despite early-stage surgical intervention potentially extending life expectancy and improving quality of life for HLHS patients, long-term efficacy is still very poor, as evidenced by very high 1-year mortality with the Norwood procedure (15% to 60%) [150]. Therefore, novel alternative therapeutic options such as stem cell therapy are urgently needed for long-term management of HLHS patients. Indeed, stem cell research has significantly advanced our understanding of human disease aetiology and a variety of stem cells have been used as an adjunctive therapy for some congenital diseases including HLHS in the past decades as discussed in the following sections. Moreover, the advent of novel stem cell technologies, such as using patient-specific iPSC and organoids, has enabled scientists to study HLHS from a new perspective which may eventually lead to better outcomes for patients.

Broadly speaking, stem cells can be divided into PSCs and adult stem cells. Adult stem cells can be further broken down into hematopoietic stem cells, mesenchymal stem cells (MSCs), and foetal adult stem cells such as umbilical cord stem cells and placental stem cells, amongst many others. Whilst many types of stem cells have been applied to study HLHS, adult stem cells have seen the greatest amount of existing scientific investigation especially in relation to clinical applications in small-scale trials [19]. In this chapter, we will explore some of the various preclinical and clinical trials that have used adult stem cells to treat HLHS and provide some updated insights into adult stem cell application in HLHS (Table 2, Figure 1).

### 3.1. Umbilical Cord Derived Stem Cells

The umbilical cord remains an excellent pre-existing supply of stem cells, consisting of umbilical cord blood mononuclear cells (UCB-MNCs) isolated from blood and mesenchymal stem cells (UCB-MSCs) extracted from Wharton’s jelly. Compared to other sources of MSCs and other adult stem cells, UCB-MSCs exhibit a better proliferative capacity, superior mesodermal differentiation potential, and a higher level of secretion of growth factors [156,157]. Preclinical investigation with UCB-MSCs in neonatal animal models resembling RV overload has seen significant potential benefits with relevance to the HLHS overloaded state. Davies et al. used neonatal ovine models receiving intramyocardial UCB-MSCs to demonstrate improvements in the RV ejection fraction (RVEF), reduced fibrosis, and increased recruitment of endogenous progenitors [158]. Importantly, data from this preclinical study also infer that the majority of benefits are attributed to the indirect effects of UCB-MSCs to enhance neovascularisation, reduce inflammation, and ultimately adverse remodelling, rather than their direct differentiation into CMs [19].

Promising preclinical data has led to translational efforts in applying UCB-MNCs in HLHS neonates. Phase I trials investigating the safety of autologous umbilical cord blood-derived cells (NCT01883076) were conducted at the Mayo Clinic with intramyocardial injection of UCB-MNCs during the Glenn procedure [153]. They were able to prove that cord cell collection, processing, and intramyocardial delivery were feasible and safe with no serious procedure-related adverse events reported. Subsequent patient follow-up assessment with measurements of RVEF showed that tricuspid valve function remained stable, further indicating no longer-term detriments [153]. Brizard et al. [154] used a novel cardioplegia solution during the Norwood operation to infuse UCB-MNCs. They reported three deaths which were deemed unrelated to therapy, with the remaining survivors demonstrating desired RV function at the time of Stage II operation. Despite preclinical studies giving us the mechanistic principle of treatment and early clinical trials being largely encouraging, some UCB-MNC specific limitations remain. Most notably, challenges may arise when collecting viable umbilical cord blood in HLHS patients who are already at increased risk of hemodynamic instability [159]. Improvements and more insights into safe collection techniques and better long-term follow-up effect are urgently required in this field.

### 3.2. Bone Marrow Stem Cells

Apart from UCB-MNCs, MSCs can also be sourced from bone marrow in the form of BM-MSCs [16]. Much like its UCB-MNCs counterpart, BM-MSCs are an attractive proposition for HLHS due to their multipotency, low immunogenicity, excellent immunomodulatory effect [160], and easy isolation and collection, as well as being safe to use in vivo [161], as evidenced by the large amount of available clinical and preclinical data and translational usage [162]. Although in vitro studies showed that BM-MSCs with manipulation are capable of differentiating into mesodermal lineages including endothelial cells [163] and CM-like cells [164], their primary therapeutic benefits do not derive from the generation of the aforementioned cells. Instead, BM-MSCs are thought to promote angiogenesis [165], reduce oxidative damage [166], inhibit tissue fibrosis, and prevent cells from apoptosis [167] via paracrine signalling or through the secretion of exosomes.

Using a porcine model, Wehman et al. showed that the epicardial injection of BM-MSCs was able to minimise cardiac hypertrophy, increase capillary density, and preserve RVEF without any serious adverse events [168]. Building upon this, Liufu et al. simulated RV pressure overload in mouse models using pulmonary artery banding (PAB). They were able to demonstrate neonatal BM-MSCs had an age-dependent effect on vascular endothelial growth factor (VEGF) secretion and paracrine activity, resulting in improved RV function [169]. Whilst providing valuable insight, this study also suggests that BM-MSCs provide meaningful benefits only if collected from very young donors, raising the inevitable ethical dilemma of using them to treat HLHS. Nevertheless, multiple research groups have conducted Phase I/II clinical trials using BM-MSCs from adult volunteers with some promising results. The ELPIS trial was able to demonstrate safety and potential benefits of intramyocardial injection of allogeneic BM-MSCs (Lomocel-B) [20], which has seen progress into the Phase IIb ELPIS II trial (NCT04925024). This trial is currently ongoing with aims to recruit 38 patients for a blinded, randomised control trial to provide more concrete evidence of efficacy compared to the open, non-randomised, single-arm ELPIS Phase I trial.

A separate Phase I/II randomised study investigated the benefits of mesenchymal precursor cells (MPCs) in patients with HLHS and borderline left ventricles. The Mesoblast MPC trial (NCT03079401) [170] utilises rexlemestrocel-L, highly purified STRO-3^+^ BM-MPCs derived from adult volunteers, to treat HLHS. In total, 10 controls and 9 BM-MPC-treated patients were followed over a period of 24 months with their cardiac parameters measured by 2D echocardiography, 3D echocardiography, cardiac MRI, and catheterisation. When indexed for body surface area, the BM-MPC group demonstrated significant improvements in multiple 2D and 3D echocardiography parameters [170], including LV end diastolic volume and end systolic volume. However, such improvements were absent in cardiac MRI findings. Due to the conflicting results and underpowered nature of the study, no definitive conclusion should be made regarding the efficacy of BM-MPCs in HLHS. Despite this, the feasibility and safety demonstrated by the study once again reinforce the appropriateness of future, more well-powered BM-MPC studies for HLHS. Interestingly, the authors also commented on the proportion and percentage of patients receiving biventricular (BiV) circulation, a procedure that significantly improves prognosis, with 5/5 BM-MPC patients having successfully completed the surgery compared to 4/7 for the control [170]. While low participant numbers limit the validity of findings, they may still help to generate hypotheses about the possibility of BM-MPC therapy in improving success rates of BiV in HLHS patients with better-powered studies in the future.

In addition to BM-MSCs, bone marrow harbours other stem cell populations including endothelial progenitor cells and hematopoietic stem cells which have been studied extensively and have been a mainstay of treatment for many haematological conditions for many years [171,172,173]. However, there is limited evidence to show any potential benefit of non-mesenchymal stem cell populations from bone marrow to HLHS patients. To date, only one case report exists amongst the available literature, in which the authors reported a 25-year-old HLHS patient receiving an intracoronary infusion of autologous BM-MNCs [174]. No serious adverse events were recorded and the patient saw significant improvement in RVEF, most notably 3 months post-infusion with an increase from 30% to 40%. This case report suggests BM-MNCs may prove to be yet another avenue of research to support RV function post-palliative surgery in HLHS patients.

### 3.3. Cardiac Stem and Cardiosphere-Derived Cells

Cardiac stem cells (CSCs) are another avenue that has been investigated for their possible use in HLHS patients. The concept for using CSCs for cardiac diseases has evolved significantly within the last two decades. Earlier efforts focused on a specific group of endogenous CSCs known as c-kit cells for the regeneration of new CMs following cardiac injury. However, multiple subsequent studies have failed to replicate c-kit cells’ cardiomyogenic potential [175,176]. This has left their exact therapeutic potential largely indeterminate and diverted research efforts into other elements of CSCs. Current CSC research is mainly focused on cardiosphere-derived cells (CDCs) which are cells derived from cardiospheres that comprise a heterogeneous population containing mesenchymal cells, c-kit cells, and endothelial cells [177]. Preclinical studies using CDCs were able to demonstrate improved cardiac remodelling in the infarcted myocardium of murine [178,179] and porcine [180,181] models. Specifically, compared to control treatment, CDCs were able to reduce scar size, improve ventricular compliance/function [179], and enhance neovascularisation [181]. Importantly, by using models mimicking right ventricular overload similar to a post-surgical HLHS state, researchers were able to show that CDC treatment could improve RVEF and reduce cardiac fibrosis, with the mechanism of benefits thought to be derived through paracrine effects [182].

Based on the promising outcome in preclinical studies with CDCs, the 2015 TICAP trial (Transcoronary Infusion of Cardiac Progenitor Cells in Patients With Single-Ventricle Physiology) was the first to utilise CDCs in the context of HLHS patients [151]. Fourteen infants due to undergo Stage II and III palliative surgery were either given CDC intracoronary therapy or control. The results showed improvements in RVEF from 46.9 ± 4.6% to 54.0 ± 2.8% at 18 months (*p* = 0.0004), with observable functional improvements in heart failure status measured by the New York University Paediatric Heart Failure Index (NYUPHFI) [151].

The later, larger, Phase II trial PERSEUS (Cardiac Progenitor Cell Infusion to Treat Univentricular Heart Disease) similarly evaluated the efficacy of intracoronary CDCs during Stage II/III palliative surgery in 34 paediatric patients [152]. Analysis at three months post-therapy demonstrated an improvement of +6.4% versus +1.3%; (*p* = 0.003) in RVEF compared to control. Subsequently, patients in the control group were given the option of receiving CDCs 4 months after surgery with all 17 patients electing to do so [152]. Similar improvement with post-CDC infusion was noted in RVEF (38.8% versus 34.8%; *p* < 0.0001) in these patients. Secondary outcomes of heart failure status, quality of life, and somatic growth also saw improvements. Follow-up analysis using combined data from TICAP and PERSEUS by Sano et al. [183] compared controls to treatment and found significantly improved ventricular function with lower rates of complications and follow-up procedures. Similarly, Tauri et al. [184] conducted a 3-year follow-up on TICAP patients and confirmed persistently improved RVEF (+8.0% versus +2.2%; *p* = 0.03) with no serious long-term complications noted. Importantly, long-term follow-up analysis of the TICAP/PERSEUS trials confirmed that CDC infusion was associated with lower hazards of late failure and adverse events, demonstrating a durable clinical benefit of CDC infusion in patients with heart failure with reduced ejection fraction over 8 years [185]. Although issues with the small sample size, heterogeneity in patient staging, and lack of blinding analysis may affect what we can extrapolate from the data, early-phase clinical trials provide sufficient evidence for more rigorous future studies that can truly investigate the therapeutic benefits of CDCs in HLHS.

## 4. PSC Derived Cardiomyocytes for Studying HLHS

### 4.1. PSC-Derived Cardiomyocytes from HLHS Patients

Since the discovery of iPSCs by Yamanaka et al. in 2006 [186], their application in CHD has yielded valuable insights into pathogenesis and novel therapeutic options [187,188]. Briefly, iPSCs are reprogrammed from somatic cells such as dermal fibroblasts harvested from a donor of interest into a pluripotent state that closely resembles that of ESCs [189]. This reprogramming is achieved via the ectopic transduction of a combination of key transcription factors (OCT4, SOX2, KLF4, and c-MYC, collectively known as OSKM) into dermal fibroblasts [189]. OSKM initiates the reversal of differentiation via epigenetic remodelling, silencing of somatic lineage gene enhancers, and activation of pluripotency-associated gene enhancers. The overall process leads to widespread changes resulting in cells with different metabolism, signalling, and nuclear architecture. Due to clonal competition and heterogeneity, the vast majority of cell colonies do not successfully transition into iPSCs [190]. Surviving and suitable iPSC colonies are selected and screened for future use by analysing their karyotype stability, pluripotency, and trilineage differentiation potential in vitro and in vivo [190].

Once the iPSCs have been validated, the next step is to differentiate into viable CMs. Currently, there is no single universally recognised and standardised protocol for iPSC-CM derivation, with different protocols being documented to generate chamber-specific CMs [21,191,192]. Multiple studies have shown that patient-specific iPSC-derived CMs could be a promising and powerful tool to study HLHS pathogenesis and the causal effects of genetic variants on HLHS phenotypes or aetiology (Table 3). For the study of HLHS, the foundational principles of iPSC-CM generation for ventricular CMs remain the most clinically relevant and involve manipulation of the Wnt/β-catenin signalling pathway [21]. Broadly speaking, in the first 24 h, iPSCs are induced into mesoderm using a Wnt agonist, most commonly CHIR99021. Following mesoderm induction, inhibition of Wnt is achieved with IWP-2, IWP-4, or IWR-1 normally around day 3–4 of CM differentiation. By doing so, mesoderm is guided towards a cardiac fate. Insulin can also be added from day 7 onwards to support cell growth and survival [193], with spontaneously contracting clusters of CMs typically appearing at this point [194]. The purity of the CM population is also enhanced with the metabolic selection of cells by culturing them in glucose-free and lactate-rich medium promoting the preferential survival of CMs that is capable of utilising oxidative metabolism [195]. Final validation of iPSC-CMs is achieved through immunostaining or flow cytometry for cardiac markers such as cTnT or a-actinin, structural imaging to assess sarcomeres, or electrophysiological assays [194].

### 4.2. Human iPSC-Derived Cardiomyocytes as Disease Modelling for HLHS

Advances in iPSC technology have enabled disease modelling at an unprecedented level, especially for CHDs such as HLHS. By reprogramming dermal fibroblasts or other cells from HLHS patients into iPSCs and subsequently differentiating them into CMs and other cardiac cells [205,206], researchers are able to model HLHS in vitro. Using the aforementioned techniques, several studies were able to demonstrate that HLHS patient-derived iPSCs displayed impaired CM differentiation, decreased proliferation capacity, changed expression levels of key cardiac markers, downregulated biological pathways related to the cell cycle, DNA replication, and DNA repair, a lower ability to give rise to beating clusters, reduced oxygen consumption rate and ATP production, disorganised sarcomere structure, different calcium transient patterns, and altered electrophysiological properties relative to controls [97,197,200,201,204,207] (Table 3). Kobayashi et al. showed that key transcription factors including HAND1/2, NKX2.5, and NOTCH-1 were significantly downregulated in HLHS iPSC-CMs [141]. Epigenetic modifications have been suggested to contribute to these genes’ downregulation in HLHS-CMs. Specifically, reduced H3K4 dimethylation and elevated H3K27 trimethylation at the NKX2.5 promoter was identified in HLHS iPSC-CMs. This is consistent with the transcriptional silencing of NKX2.5 and downregulation of cardiac-specific marker genes such as TNNT2 and NPPA [141]. In addition, HLHS iPSC-CMs showed reduced contractility and altered calcium handling signalling, suggesting an intrinsic CM defect as a key part of HLHS pathogenesis rather than solely as a consequence of abnormal blood flow during foetal development. However, caution should be taken when interpreting the results reported by Kobayashi et al. [141], since this study does not give any information about the phenotype of the HLHS patients and it is very difficult to tell if the HLHS iPSCs and associated data reported in this study are generated from one or several patients.

Later studies such as Yang et al. derived iPSCs from five HLHS patients to further investigate NOTCH signalling pathway defects as the driver of HLHS pathogenesis [200]. Exome sequencing using patient fibroblasts highlighted significant deletions in multiple aspects of the NOTCH pathway. iPSCs from HLHS patients displayed multiple abnormalities of the NOTCH pathway including reduced NOTCH receptors and downstream target effects, diminished generation of cardiac progenitor cells, a decreased CM contraction rate, and an increased propensity towards smooth muscle compared to control iPSCs [200]. Reversal of defective NOTCH signalling using a Jagged-1 peptide, a NOTCH ligand, was critically able to restore normal CM functions by improving sarcomere organisation and beating frequency [200]. These results cement the role of dysfunctional NOTCH signalling as a key driver of pathogenesis in HLHS.

More recently, Ye et al. [198] employed CRISPR/Cas9 to generate NOTCH1 knockout iPSCs and demonstrated that NOTCH1 deficiency disrupted cardiac progenitor differentiation. NOTCH1-knockout iPSCs displayed a significant preference towards atrial lineage differentiation with disrupted ventricular lineage differentiation and suppressed proliferation of CMs via downregulation of NOTCH and WNT downstream effects. Single-cell RNA sequencing revealed NOTCH1’s critical role in cardiac mesoderm transformation into FHF which eventually contributed to left ventricle formation, thus providing an additional explanation for the ventricular hypoplasia observed in HLHS in addition to the “no flow, no grow” model as mentioned in the above section.

Using HLHS iPSC-CM models, researchers were also able to reveal that NOTCH pathway defects may be amplified by dysregulation of nitric oxide (NO)-dependent signalling during early cardiogenesis [202]. HLHS iPSC-CMs consistently displayed lower levels of NO and decreased activation of NICD, contributing to defective early cardiac lineage differentiation. Such a linkage is further evidenced by the partial restoration of differentiation capability and CM maturation with the supplementation of NO.

Altogether, these iPSC-based studies provide compelling evidence that HLHS is driven by intrinsic defects in cardiogenesis and dysregulation of signalling pathways such as NOTCH1, NKX2-5, and HAND1 (Table 3). The use of iPSC-CM models has allowed us to greatly expand on the previously dominant theory of hemodynamic compromise as the main driver of pathogenesis in HLHS into that of molecular and genetic defects and key factors of the disease. Better understanding of disease modelling and aetiology will guide future therapies and even the possibility of personalised therapeutic strategies.

### 4.3. HLHS iPSC-CMs: A Platform for Drug Discovery and Evaluating Drug Toxicity

Recent progress in identifying growth factors, transcription factors, and signalling pathways critical to HLHS pathogenesis has also established potential therapeutic targets. Patient-specific iPSC-derived CMs have been used as a promising platform to confirm and identify potential therapeutics for HLHS (Table 3). In addition to the use of NO [202] and Jagged-1 [36] to rescue NOTCH signalling defects, other specific targets for HLHS have also been studied as a potential therapeutic option. Paige et al. [204] used iPSCs from HLHS patients with the MYH6-R443P variant to show the deficient contraction force and velocity of HLHS iPSC-CMs. Subsequent correction of the gene mutation was able to effectively improve contraction and velocity in the iPSC-CMs. Xu et al. [59] also reported an increase in apoptosis and oxidative stress in HLHS iPSC-CMs, features that are frequently observed with early-onset heart failure in clinical practice. Importantly, an observable reduction in apoptosis and oxidative stress was reported with sildenafil and tauroursodeoxycholic acid treatment. In addition, partial improvements in mitochondrial and contractile defects alongside enhanced fatty acid supplementation and improved sarcomere function were noted. These studies highlight the possibility of utilising iPSCs in identifying potential cardioprotective compounds in HLHS patients.

Beyond identifying therapeutic targets, iPSCs have also been proposed as a tool for evaluating drug toxicity in many organs, in particular for the heart [208,209,210]. Compared to traditional animal models and other in vitro systems in drug testing and toxicity assessment, iPSCs offer several distinct advantages. The most important of which is direct generation from human donors which allows for models that most accurately reflect human physiology and disease-specific phenotypes, thereby enhancing the predictive value of preclinical testing. The scalability of iPSCs also makes them highly compatible with high-throughput screening technologies which allows for rapid drug evaluation with easy-to-measure endpoints. Additional benefits with iPSC-CMs may also come from reduced ethical concerns and fewer regulatory barriers compared to animal model alternatives.

Currently, one area of cardiology which has seen a strong promotion for the integration of iPSC-based drug testing and toxicity is the study of arrhythmogenic drugs. This is highlighted by the FDA and counterparts in Japan and Europe devising the Comprehensive In vitro Proarrhythmia Assay (CiPA) using iPSC-CMs to study drugs with a high propensity of causing torsades des pointes (tDP) [211,212,213]. Improvements made in high-throughput screening (HTS) by incorporating iPSC-CMs have seen major advances to current and future candidate drug identification and toxicity screening. Briefly, HTS is a highly automated system that allows the rapid testing of hundreds to thousands of candidate compounds simultaneously. Miniature assays using iPSC-CMs alongside the tested compound can provide quantifiable readouts by measuring absorbance or fluorescence to measure response [214]. Compounds that produce the desired change in fluorescence or absorbance are then validated in this high-efficiency system to identify bioactive molecules or toxicity. Overall, HTS leads to a significantly accelerated early-phase drug testing system. The integration of iPSC-CMs within HTS has shown huge promise. For instance, Huang et al. [215] generated iPSC-CMs from 13 HLA-super donors that could represent a significant proportion of the world population. They subsequently used an HTS to test a wide range of compounds with these iPSC-CMs and were able to demonstrate dose-dependent toxicity amongst five previously known cardiotoxic drugs and a further four previously undescribed drugs. Extensive electrophysiological studies and further in vivo studies were able to demonstrate a significant adverse effect on CMs caused by these drugs, thus demonstrating the utility of iPSC-CMs in identifying potential drug toxicities during the preclinical stage.

In the context of HLHS, iPSC-CMs from patients retain patient-specific genetic and epigenetic signatures that allow for the study of drug toxicity in a precision medicine approach. Certain compounds tested may exacerbate or alleviate pathological features in some subsets of HLHS whilst having little effect on other subsets. HTS platforms employing HLHS-specific iPSC-CMs allow for rapid evaluation of candidate drugs by measuring their potential effects on endpoints such as electrophysiological stability, viability, and mitochondrial functions. For the case of HLHS where non-surgical options are limited with systemic complications frequently arising from lifelong haemodynamic stress, novel medications discovered using HLHS iPSC-based HTS could provide safe, effective, and tailored pharmacological interventions to HLHS individuals.

### 4.4. Limitations of iPSC-CMs 2D Model

Although the advent of iPSC-CMs has seen great strides in the aetiology and therapeutic target identification of HLHS, some limitations remain which must be acknowledged. It is crucial to note that almost all HLHS tissue from patients comes from stages where HLHS is already present, making it very difficult to know what happened early in the development of the malformation(s) associated with HLHS. Even more crucially, current studies are mainly focused on iPSC-derived CMs, potentially overlooking other important pathological causes underling HLHS aetiology. Although growing evidence has suggested a critical role for the intrinsic defects of CM proliferation and endocardial growth in HLHS aetiology [200,216], other studies also indicate that additional factors affecting the hemodynamic loading of the left heart in the foetal period can directly contribute to the pathogenesis of HLHS [34,217]. The latter studies suggest that an abnormality in the developing mitral and/or aortic valves may be the primary insult for the LV underdevelopment observed in HLHS patients, which is well-aligned with the widely supported “no flow, no grow” hypothesis. Therefore, the aetiology of HLHS is likely multifactorial, and the potential intrinsic responses of other cardiac cells such as valvular interstitial cells, cardiac endothelial cells, smooth muscle cells, and fibroblasts in the context of HLHS warrants further investigation. Importantly, although the HLHS iPSC-CM model used for drug screening and pharmaceutical interventions could recapitulate some key pathological HLHS phenotypes, it is not equivalent to HLHS patients and requires further optimisation and improvement.

Another of the most frequently cited limitations is the immaturity of iPSC-CMs, characterised by structural and functional differences, such as underdeveloped sarcomeres, poor calcium handling and energy metabolism, and an immature electrophysiological profile [218]. Immature iPSC-CMs are more akin to the foetal heart with different organisation, calcium handling, electrophysiology, and metabolism which do not translate directly to the high-pressure, post-natal physiology seen in HLHS patients [219,220,221]. This directly applies to any attempts at studying the translational potential of iPSC-CM-based techniques such as cardiac patches. Whilst preliminary data may be positive, more comprehensive analysis into the clinical outcomes will be needed to ensure that iPSC-CM-based therapies can integrate effectively without significant adverse effects of arrhythmias, tumorigenesis, and immune rejection. Attempts at addressing the issue of iPSC-CM immaturity have already seen fruitful efforts by groups trying a variety of methods. One possible way to achieve more mature CMs is to apply mechanical stretching alongside electrical stimulation to iPSC-CMs [222,223]. The resulting iPSC-CMs exhibited better calcium handling ability, increased mitochondrial density, improved sarcomere length and organisation, and a gene expression profile more akin to adult CMs. Moreover, T-tubule formation alongside enhanced calcium release and improved excitation–contraction coupling were observed in iPSC-CMs treated with hormones such as triiodothyronine and dexamethasone [224]. Furthermore, metabolic strategies using fatty acid supplementation have also been shown to induce a shift from glycolysis to oxidative phosphorylation alongside improving sarcomere organisation in iPSC-CMs [225]. The more recent literature suggests the use of multiple methods of maturation including metabolic medium optimisation, electrostimulation, and culturing in nano-patterned substrates can act synergistically to substantially improve maturation [218]. Collectively, these strategies for maturing iPSC-CMs can be used to generate more physiologically relevant cardiac models to complement existing iPSC-CM studies for HLHS and beyond.

Another complex issue posed by the current understanding of HLHS is the multifactorial nature and associations with the development of disease. Extensive genetic heterogeneity, copy number variants [226], chromosomal abnormalities and even maternal [227], and environmental impacts [228] have been attributed as risk factors for HLHS. Many studies using iPSC-CMs fail to capture these complex interactions and focus more on specific genetic or signalling pathway defects. Finally, current iPSC-CMs models often simplify the complex nature of cardiac development with a predominant focus on CMs whilst neglecting the role of endothelial, valve, and vascular smooth muscle cells in cardiogenesis.

## 5. Implications of HiPSC-Derived 3D Cardiac Patches for Treating HLHS

Three-dimensional cardiac patches or engineered heart tissues (EHTs) offer an alternative way in which iPSC can be used for modelling HLHS and exploring potential future therapies. Briefly, cardiac patches are generated initially using iPSC or other stem cells, and they are then differentiated into the desired cell types such as CMs, endothelial cells, and cardiac fibroblasts. These cells are seeded onto a biomaterial scaffold that supports growth and is designed to mimic the extracellular matrix (ECM) [229]. The scaffold can either be entirely synthetic made from polymers such as polyglycolic acid (PGA) and polycaprolactone (PCL) or from decellularised ECM [230]. Once seeded, the patch is cultured in a bioreactor to promote tissue maturation and form a functional cardiac patch (Figure 2A).

Compared to 2D monolayers of iPSC-CMs which are unable to replicate complex structural defects, 3D cardiac patches derived from iPSCs provide a more physiologically relevant model of HLHS [231]. Krane et al. [201] was able to utilise 3D heart patches derived from iPSCs to investigate deeper underlying intrinsic defects associated with HLHS (Figure 2B). Corroborating evidence from 2D models, 3D cardiac patches generated from HLHS iPSCs were able to demonstrate a significant increase in MLC2a+ cells and a reduction in MLC2v+ cells representing atrial and ventricular cells, respectively, thus indicating a poor maturation response to growth [201]. Crucially, HLHS iPSC-derived 3D cardiac patches were able to uncover additional evidence of DNA damage and CM apoptosis, with roughly 50 percent of CMs demonstrating three or more nuclei compared to roughly 5 percent in control cardiac patches. By looking at cell cycle markers of Ki67 for active cycling and PH3 for mitosis, researchers found that although more HLHS cells were attempting to divide, few cells actually completed mitosis, indicating a severe disruption in the cell cycle. Further investigation showed a high correlation between multinucleated cells with increased apoptosis, as these cells were much more likely to be positive for TUNEL and cleaved caspase-3 both indicating DNA damage and programmed cell death [201]. Together, these results signify a strong genetic and cellular basis of HLHS occurring independent of hemodynamic factors. Furthermore, despite the heterogeneity of HLHS, this study was able to demonstrate a convergence of de novo mutations resulting in aberrant key pathways linked to cardiac development, chromatin organisation, and cell cycle phases [201]. These data highlight that potential therapeutic targets may be better identified by investigating compounds that can correct common dysregulated pathways rather than focusing on individual target mutations.

In the field of cardiac surgery, conventional cardiac patches made from bovine pericardium or synthetic materials such as polyester are already incorporated into clinical practice for the repair of many CHDs including HLHS [232]. Compared to conventional cardiac patches, cardiac patches derived from iPSC-CMs are able to integrate into native tissue, conduct impulses, and promote revascularisation to damaged areas of the heart [233,234,235]. Importantly, they could eventually degrade, hence eliminating the need for multiple reinterventions and reducing the likelihood of fibrosis and calcification [236]. Many preclinical trials have investigated cardiac patches as a potential therapeutic option in particular for ischemic heart disease and heart failure. Heart failure in particular is a relevant complication of HLHS. Although surgical advances have seen a significant improvement in mortality for HLHS, the resulting single RV is not well adapted for its new role of long-term systemic circulation, leading to the need for a heart transplant secondary to RV failure. In fact, alongside arrhythmias, heart failure is the leading cause of morbidity amongst HLHS patients who have had surgical repair [237]. Although cardiac patches are mainly used for repairing infarcted cardiac tissues and its therapeutic potential in HLHS has not been explored yet, iPSC-CM cardiac patches could be useful in repairing developmental defects of the atrial septum and mitral or aortic valve, as well as in supporting the hypoplastic left heart.

Recent studies in large and small animal models have highlighted potential iPSC-CM-based cardiac patches as a way to remuscularise infarcted myocardium. iPSC-CM-based cardiac patches are able to successfully integrate within the host structure and restore cardiac contractile and physiological functions in animal models [233,234]. Building on these initial studies, Querdal et al. was able to establish a dose-dependent relationship between cardiac patches and functional recovery in guinea pig models, hence further validating the therapeutic value of cardiac patches [238]. Other studies were able to confirm similar benefits in their respective models, with no significant adverse effects such as arrhythmias or tumorigenesis observed [236,239]. In the context of HLHS in which RV failure remains a major issue, iPSC-CM cardiac patches offer a promising avenue to explore for supporting the overworked RV myocardium or even as a possibility of repairing the hypoplastic LV.

Aside from preclinical trials, large-scale clinical trial data for the implementation of iPSC-based cardiac patches have yet to be published. Nevertheless, relevant clinical trials have been actively recruiting such as the bioVAT-HF Phase I/II trial (NCT04396899). Based on preclinical validation on rats and rhesus macaques, the study aims to prove the hypothesis that engineered heart muscle (EHM), a form of cardiac patch derived from stromal cells and iPSC-CMs, can be safely and effectively implemented to improve cardiac function in heart failure patients. Thus far, the study conducted by Zimmerman et al. has been recruiting patients with advanced heart failure with an ejection fraction less than 35% and no realistic option of heart transplant. A significant recent update published by the group shows a proof of concept for the use of EHM in humans. This is demonstrated by the study of a patient’s heart with EHM implanted after the patient had received a successful heart transplant. The explanted heart was shown to contain viable human CMs that were vascularised and successful integration of the EHM with no signs of adverse outcomes such as tumour formation, immune rejection, or arrhythmias [234]. Although wider implications of a significant therapeutic benefit cannot be derived from a single case, the examined evidence does support the continuation of the broader clinical trial. This trial alongside other early implementations of cardiac patches in isolated case reports in humans highlights the feasibility of using cardiac patches as a part of the larger HLHS therapy and in particular in the case of post-surgical RV failure for HLHS patients. Further studies in HLHS-specific preclinical models will be required before clinical translation can be safely applied.

## 6. Future Directions

Future studies in iPSC technology for HLHS will need to address these critiques in order to improve our understanding of the disease and therapeutic options. With the advances of 3D bioengineering and innovation in stem cell technology such as organoids [240,241], scientists are finding innovative ways to make iPSC-based models and therapies ever more clinically applicable.

### 6.1. Cardiac Organoids: A Better Model for HLHS?

As previously explored, HLHS consists of the underdevelopment of multiple structures of the left heart including the LV, mitral valve, aortic valve, and ascending aorta, alongside genetic and cellular defects. While 2D iPSC-derived models and traditional animal models have previously given us much insight into HLHS pathogenesis, they fail to capture the complex and variable nature of HLHS. Recent developments have seen the emergence of cardiac organoids (COs) or cardioids which are used to advance our understanding of CHDs including HLHS. COs are self-organising 3D structures derived from PSCs (iPSCs and ESCs) [196,210]. Multiple experimental protocols have been reported to generate COs from human PSCs (Figure 3). They consist of CMs, endothelial cells, and fibroblasts that aggregate together to better mimic human cardiac tissue, architecture, and functional dynamics [196,242]. Interestingly, some PSC-derived COs could recapitulate patterns of early cardiomyogenesis, resembling early embryonic heart anlagen and nascent foregut endoderm [243]. Once generated, these COs are capable of spontaneous rhythmic contractions, with some studies even demonstrating the ability to derive vascularised and chambered COs [244]. These features combined make cardioids excellent vehicles to study CHDs such as HLHS.

Much like their 2D counterparts in iPSC-CMs, COs have been used as a method of modelling HLHS to uncover even more insights into the aetiology of HLHS. One landmark study by Hofbauer et al. [196] highlights how 3D self-organising cardioid models can advance our understanding of pathogenesis beyond the insight provided by 2D iPSC models alone. Focusing on the role of HAND1, a transcription factor implicated in left ventricular development, the study used CRISPR-Cas9 mediated knockout of HAND1 in iPSC to evaluate its role during cardioid formation. They were able to observe several key differences between HAND1-KO cardioids and control cardioids. Although HAND1-KO cardioids were capable of specification into TNNT2^+^ CMs, they consistently either failed or formed significantly smaller, disorganised cavities. These findings indicate that HAND1 is essential for chamber morphogenesis but not for cardiac lineage commitment [196].

Parallel knockout experiments for NKX2.5 in cardioids were also conducted and did not show any cardiac cavity abnormalities or any effect on HAND1 levels during the mesodermal stage. However, HAND1 levels in NKX2.5 KO cardioids were significantly reduced in the CM stage, suggesting that HAND1 operates downstream of NKX2.5 in a stage-specific manner during cardiac development [196]. Subsequently, the authors investigated the upstream regulation of HAND1 by manipulating levels of BMP to assess its impacts on downstream signalling pathways. Increasing BMP activity led to increased HAND1 expression and improved cavity formation and vice versa when BMP levels were reduced [196]. This establishes the critical importance of the BMP–HAND1 axis in controlling chamber formation. As a whole, the cardioid model in this study proved to be a powerful tool for dissecting the molecular basis of cardiac morphogenesis and enabling a precise spatiotemporal understanding of gene function and signalling pathways that a 2D iPSC-CM model simply cannot replicate (Figure 3).

Alternative protocols for generating COs have also been used to better understand early human cardiogenesis. Drakhlis et al. [243] used a protocol to generate COs from human PSCs embedded within Matrigel. The generated COs contained complex structures encompassing foregut endoderm tissue and vascular tissue alongside cardiac cells which they called heart-forming organoids (HFOs). HFOs with knockout of NKX2.5, a gene heavily implicated in HLHS, exhibited a much less compact cardiac tissue and a larger size alongside a greater proportion of disorganised sarcomeres [243]. In addition, the authors reported significantly altered gene expression profile with an increase in markers for smooth muscle differentiation, whilst genes associated with normal cardiac development such as IRX4 and NPPA were downregulated in NKX2.5-knockout HFOs, suggesting a deviation from normal cardiac cell fate [243]. Although not specifically designed with HLHS modelling in mind, the results suggest the possibility of using 3D CO models to further our understanding of key genes in HLHS, potentially acting as a powerful tool for future investigation of molecular and genetic basis of HLHS.

Beyond HLHS, COs have been employed for the study of congenital cardiac conditions. For instance, in the study by Lewis-Israeli et al. [242], COs were used to study how maternal risk factors such as pregestational diabetes may affect cardiogenesis and lead to CHD. Other common cardiac conditions such myocardial infarcts have also been successfully replicated using CO-based models, thus providing a robust alternative to animal models [245,246,247]. Aside from disease modelling, COs have been evaluated for their use in drug toxicity as well as screening therapeutic compounds [210,246]. It is highly plausible to assume HLHS COs will be utilised even further for future studies of HLHS pathogenesis and therapeutics.

Although 3D cardiac patches and COs have already demonstrated significant promise, one possible future research direction could include the use of patient-derived iPSC-COs to investigate HLHS-associated genes. Whilst current 3D cardiac patches and COs are already much more complex than 2D traditional models, to move beyond primitive heart modelling will require even more complex organoids that incorporate anatomically accurate atria, ventricles, conduction systems, immune components, and neural crest integration (Figure 4A). Interestingly, a recent study showed that the addition of 2 μM CHIR99021 during the first 2 days of CO formation and transient 4-day addition of 3 μM DY131 combined with 10 μM MK8722 (days 24–28) significantly improved CO maturation, enabling complex disease modelling and drug discovery [248]. The creation of fully developed, matured, and chambered COs or 3D cardiac patches that closely resemble HLHS will be essential to better mimic the structural and cellular complexity of the human heart and further our understanding. Potential incorporation of highly accurate HLHS COs into body-on-chip systems coupled with pumps to allow for high-pressure microfluidics may also allow us to better study the hemodynamic challenges that can arise in HLHS. Beyond disease modelling, enhanced COs platforms and improved 3D cardiac patches may facilitate more applications for HLHS such as high-throughput drug screening and the eventual development of personalised regenerative medicine strategies for HLHS individuals (Figure 4B).

### 6.2. Three-Dimensional/Four-Dimensional Bioprinting PSC-Derived Cardiac Tissues for Studying HLHS

Three-dimensional bioprinting is another innovative three-dimensional strategy which can potentially provide high-fidelity models to study HLHS. The idea of 3D bioprinting tissues builds on advances in 3D printing itself, by replacing commonly used materials such as thermoplastic with cells, often derived from stem cells alongside biomaterials and growth factors to create alternative 3D tissue constructs [249]. Currently, multiple approaches have been reliably described for the creation of 3D-bioprinted tissue structures [250]. Future research could prioritise systemic optimisation and standardisation of protocols to enhance reproducibility and to enable a more seamless transition into clinical applications. Compared to 2D models which require external maturation, 3D bioprinting using the appropriate ECM-like hydrogel can produce CMs with proven features of maturation. Three-dimensional bioprinted cardiac tissues displayed spontaneous contractility with organised sarcomeres alongside increased expression of multiple cardiac-specific markers such as MYL2, MYH7, TNNI3, and KCNJ4 [26], supporting the use of 3D bioprinting in HLHS and beyond. Currently, most of the research conducted using 3D bioprinting in HLHS comes from stem cells derived from healthy volunteers. Once the protocol for generating 3D-bioprinted cardiac structures becomes even more well-established, the next logical step would be to generate 3D-bioprinted cardiac tissue constructs using HLHS-specific iPSCs. Such a movement will allow us to study the morphogenetic disruptions seen in HLHS and identify novel therapeutic compounds for HLHS. Although 3D bioprinting of the whole heart remains some way away, smaller tissue constructs such as valves, vessels, and cardiac patches may be more precisely recreated than traditional seeding and scaffolding techniques. Based on the specific developmental defects of patients with HLHS, personalised cardiac patches and small piece of aortic or mitral valves, as well as aorta, could be precisely generated using a 3D-bioprinting technique, which can potentially be used for repairing the developmental defects, as well as supporting a hypoplastic LV or overworked RV in HLHS individuals.

Concurrently, scientists have been taking inspiration from the latest advances in material sciences to hypothesise about potential 4D-bioprinting tissue techniques [251]. Currently a 3D-bioprinted tissue structure remains a static and relatively unadaptable piece of tissue. The properties of some smart biomaterials allow them to change characteristics such as shape, size, and elasticity in response to a variety of stimuli such as pH, mechanical force, and electrical stimulation [252] (Figure 5). By engineering heart tissue integrating smart materials, we may be able to produce 4D-bioprinted valves, cardiac patches, or even whole hearts in the long term that are capable of “growing” and adapting with changes to their environment. This technology would allow much better integration into host tissue and reduce the need for surgical revisions that a 3D-bioprinted structure would likely require, representing an excellent option for paediatric surgery, particularly HLHS.

## 7. Conclusions

Surgical advances have greatly improved mortality rates in HLHS, yet long-term complications remain a major concern. Despite extensive research, current understanding of HLHS pathophysiology remains incomplete with multiple factors implicated in contributing to a highly heterogeneous presentation of this severe and complex CHD. Stem cell-based techniques have provided an alternative avenue to study HLHS pathogenesis by uncovering complex, intrinsic molecular-level transcriptional defects in HLHS cardiomyocytes. Early-phase clinical trials using adult stem cell populations have proven safety and feasibility with ongoing studies investigating potential therapeutic benefits. Meanwhile, emerging stem cell techniques in the form of COs and multi-dimensional bioprinting tissue constructs have provided us with ever more sophisticated high-fidelity models that recapitulate key disease features. These advances not only deepens our mechanistic understanding of HLHS but also provide robust platforms which can be used for high-throughput drug screening and the development of personalised regenerative medicine strategies in the near future.

## Figures and Tables

**Figure 1 cells-14-01396-f001:**
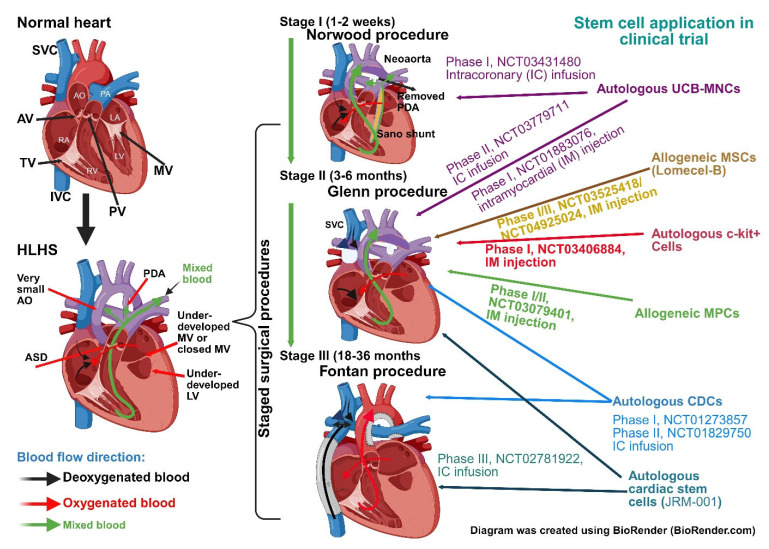
Cardiac developmental abnormalities, staged surgical procedures, and representative stem cell treatments in hypoplastic left heart syndrome (HLHS). The key cardiac developmental abnormalities of HLHS include ASD (atrial septal defect), PDA (patent ductus arteriosus), very small aorta (AO), significant underdeveloped LV (left ventricles), and underdeveloped MV (mitral valve) or closed MV, resulting in ineffective systemic circulation and early immature mortality. Multiple surgical interventions need to be conducted at early stages (Norwood at 1–2 weeks, Glenn at 3–6 months, and Fontan at 18–36 months) to establish and restore ineffective systemic circulation and prevent early immature death. Based on promising findings from preclinical studies using stem cell therapy in HLHS, a series of clinical trials were conducted to evaluate the feasibility, safety, and therapeutic potential of different adult stem cells in HLHS patients. Stem cells were administered to patients at different stages of surgical procedures. Findings from most clinical studies have demonstrated the good feasibility and safety of stem cell application in HLHS patients, with varied therapeutic effects being reported. AV, aortic valve; CDCs, cardiosphere-derived cells; IVC, inferior vena cava; LA, left atrium; MPCs, mesenchymal precursor cells; MSCs, mesenchymal stem cells; MV, mitral valve; PA, pulmonary artery; PV, pulmonary valve; RA, right atrium; RV, right ventricle; SVC, superior vena cava; TV, tricuspid valve; UCB-MNCs, umbilical cord blood mononuclear cells.

**Figure 2 cells-14-01396-f002:**
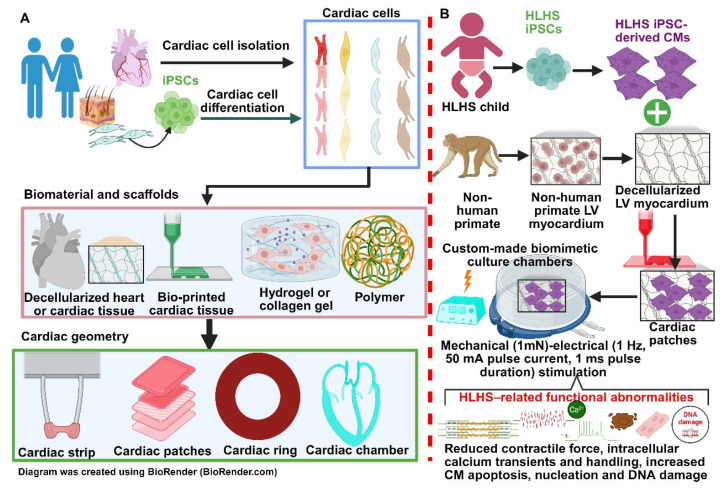
Human iPSC-derived 3D cardiac patches and applications in HLHS. (**A**) Multiple types of 3D cardiac structures (cardiac strip, patches, ring, and chamber) are generated from patients through several strategies by using cardiac cells (cardiomyocyte, smooth muscle cell, endothelial cell, and cardiac fibroblast) either directly isolated from heart or differentiated from patient-specific iPSCs with different biomaterial and techniques (decellularised or 3D-bioprinted cardiac tissue, hydrogel/collagen gel and polymer). (**B**) Krane et al. [201] generated HLHS-specific 3D cardiac patches using HLHS iPSC-derived cardiomyocytes (CMs) and decellularised non-human primates’ left ventricular (LV) myocardium. They reported that these cardiac patches exhibited several HLHS-related functional abnormalities (impaired contractility and calcium handling and increased CM apoptosis), demonstrating that these cardiac patches are a powerful platform to study HLHS pathogenesis and identify potential therapeutics for HLHS patients.

**Figure 3 cells-14-01396-f003:**
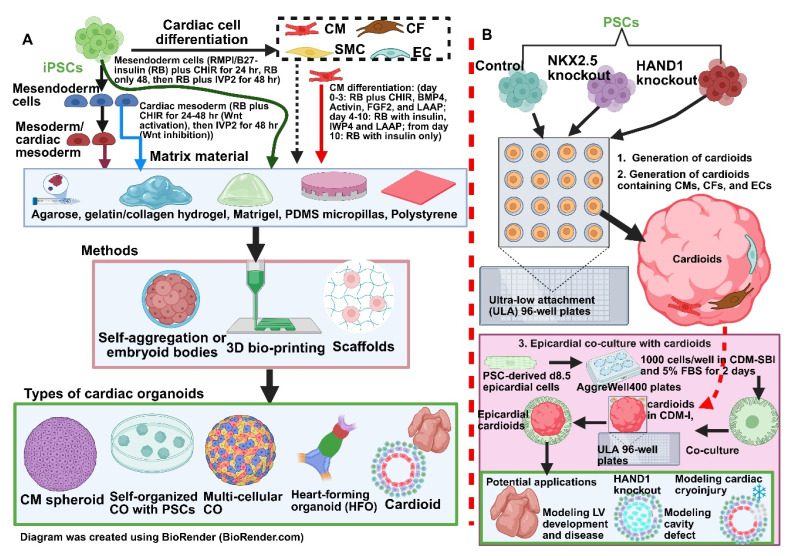
Human iPSC-derived cardiac organoids (COs) and applications in HLHS. (**A**) Cardiac organoids (cardiomyocyte (CM) spheroid, self-organised COs, multi-cellular COs, heart-forming organoid (HFO), and cardioid) can be generated from induced pluripotent stem cells (iPSCs) using different strategies. Human iPSCs can be differentiated into different developmental stages with the indicated protocol before CO generation, such as mesendoderm cells, mesoderm/cardiac mesoderm cells, and cardiac cells (CM, cardiac fibroblast (CF), smooth muscle cell (SMC), and endothelial cell (EC)). Human iPSCs or their derivatives (CMs with or without CFs, SMCs, and ECs) can be used to generate different COs through self-aggregation or embryoid body formation, 3D bioprinting, or scaffolds with different material (agarose, gelatin/collagen hydrogel, Matrigel, PDMS micropillars, and polystyrene). (**B**) Hofbauer et al. [196] generated PSC-derived cardioids with different cellular compositions using three experimental protocols (general cardioid, cardioids containing multiple cardiac cells, and epicardial cardioids). Combined with NKX2.5 or HAND1 gene knockout and BMP signalling inhibition, they demonstrated these cardioids could be used for studying cardiogenesis and modelling left ventricular (LV) development, cardiac cavity defects, and cardiac cryoinjury.

**Figure 4 cells-14-01396-f004:**
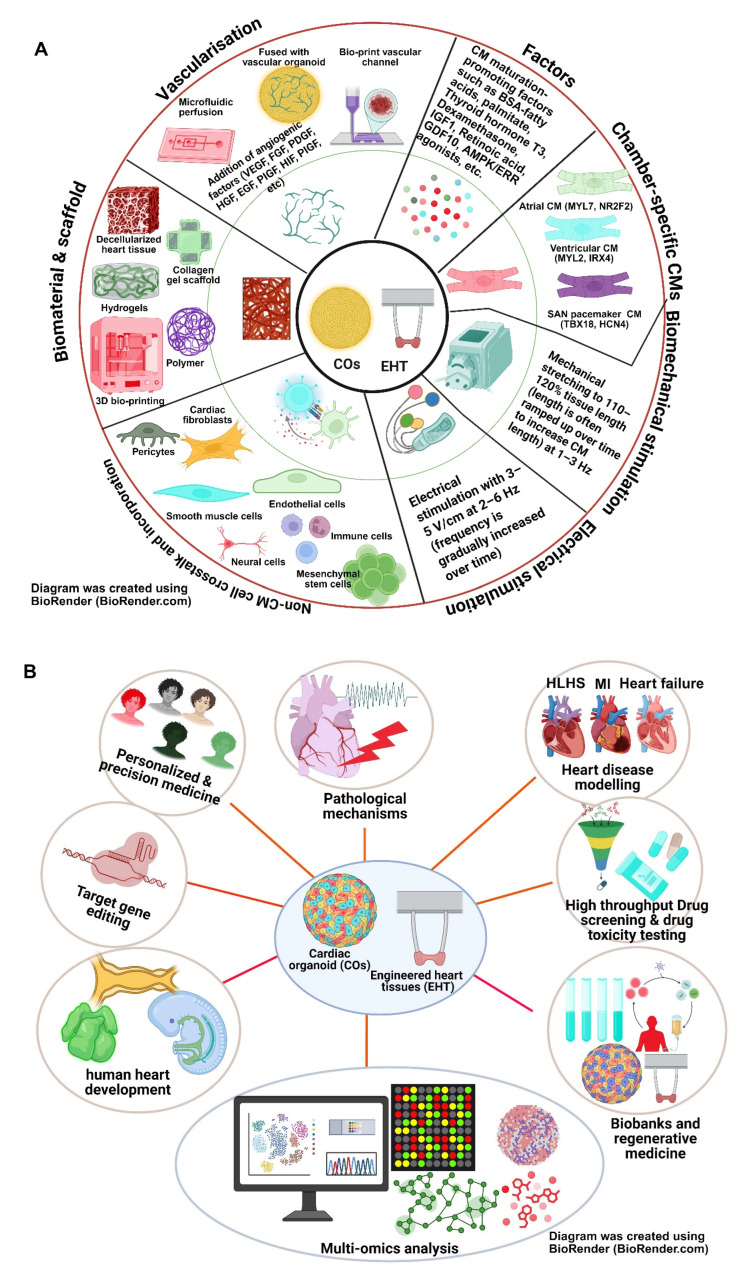
Application of cardiac organoids (COs) and potential improvements of COs and 3D engineered heart tissue (EHT). (**A**) Multiple strategies could be used to further improve CO and EHT maturation and functionalities, such as vascularisation, cardiomyocyte (CM) maturation-promoting factors, generation of chamber-specific CMs and their incorporation into COs and EHTs, biomechanical and electrical stimulation, and the incorporation of non-CM cardiac cells including fibroblasts, pericytes, smooth muscle cells, endothelial cells, immune cells, mesenchymal stem cells, and neural cells into COs and EHTs, as well as using different biomaterials and scaffolds. (**B**) Potential applications of PSC (pluripotent stem cell)-derived COs and EHTs. Increasing numbers of studies demonstrate a variety of promising applications for PSC-derived COs and EHTs, such as investigation of pathological causes of cardiac diseases, in vitro cardiac disease modelling, high-throughput drug screening and drug toxicity testing, creating human CO/EHT biobanks for regenerative medicine, multi-omics analysis to probe for novel insights into signalling pathways underlying cardiac development and disease aetiology, exploring key developmental events in human heart development, genetic engineering and editing for inherited cardiac disorders, and personalised and precision medicine for patients with cardiac diseases.

**Figure 5 cells-14-01396-f005:**
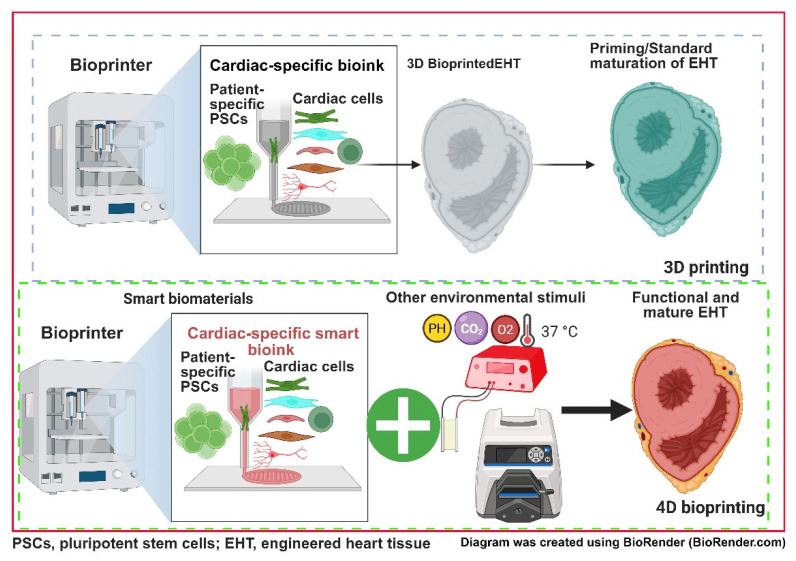
Generation of functional cardiac patches or engineered heart tissue (EHT) with 3D and 4D bioprinting. Cardiac patches and EHT can be generated through a 3D-bioprinting technique using cardiac-specific bioink and patient-specific pluripotent stem cells (PSCs) or PSC-derived cardiac cells (cardiomyocytes, fibroblasts, endothelial cells, smooth muscle cells, immune cells, and neural cells). Three-dimensional bioprinted EHTs require further improvement to truly recapitulate the anatomical geometry and functionalities of their in vivo counterparts. In this aspect, one of the newly developed strategies, 4D bioprinting, which is mainly involved in integrating smart materials into a 3D-bioprinting structure to acquire new shapes or functional properties, could be considered. After 3D bioprinting, EHT could be triggered by a stimulus (change in PH, CO_2_, O_2_, and temperature) that will self-reconfigure the structure into functional cardiac tissue.

**Table 1 cells-14-01396-t001:** Key signalling molecules, transcriptional factors, and regulatory proteins underpinning normal cardiac development and HLHS aetiology.

Signalling Pathway	Normal Function	Disease Secondary to Defect	Association with HLHS
NOTCH1	Valve formation. Ventricular septation. Left–right patterning. Regulation of progenitor differentiation [36].	Bicuspid aortic valve. Right ventricular hypoplasia. VSD [37].	Strong: Rare genetic variants (G661S, R1279H, A683T) and de novo mutations associated with HLHS found in patients and relatives [38,39]. Targeted and whole exome sequencing analysis revealed an association between a novel germline frameshift/stop-gain mutation in NOTCH1 and HLHS [40].
Bone Morphogenetic Protein (BMP)	Mesoderm induction and regulation [41]. FHF formation. Proliferation of CMs. Development of cardiac cushion [42].	Pulmonary arterial hypertension. AV canal defects [43].	Nill or weak: Although genetic animal study showed that Bmp2/4 and Bmp4/7 play a potential role in ventricular septal defects [44] and OFT septation [45], respectively, human study failed to confirm a causal relationship between BMP2/4 gene mutations with ASD, VSD, and complex CHD [46].
Retinoic Acid (RA)	Anterior–posterior patterning. FHF and SHF development. Mesoderm formation and induction [47].	DiGeorge syndrome [48]. AV cushion defects. Truncus arteriosus [49]. heart morphogenesis (posterior chamber developmental impairment and OFT septation defects) [50].	No causal genetic association between RA signalling and HLHS was reported.
Wnt/Beta-catenin	Mesoderm induction. CM differentiation. SHF expansion and patterning [51].	VSD. Truncus arteriosus. ASD. Familial exudative vitreoretinopathy. Arrhythmogenic cardiomyopathy [52,53].	HLHS family-based WGS, variant filtering, and transcriptional profiling identified 10 candidate genes including LRP2 (p.N3205D and p.A57V), and data from multi-disciplinary platforms confirmed that LRP2 is required for cardiomyocyte proliferation and differentiation [54].
Sonic Hedgehog Pathway (SHH)	Heart tube development. SHF proliferation, essential for SHF and OFT development which contribute to normal development of LV [55].	Compromised DMP formation and AVSD [53]. OFT defect [56].	No causal genetic association between RA signalling and HLHS was reported.
Hippo Pathway	Organ size. Regulates myocardial thickness [57].	Abnormal heart size [58].	HLHS iPSC-CMs showed defects in YAP-regulated antioxidant response which is associated with heart failure outcome [59].
Fibroblast Growth Factor (FGF) Family	Mesoderm induction. Outflow tracts development. SHF proliferation [60].	OFT defects. Overriding aorta [61].	Although dysregulated FGF signal pathways were observed in HLHS foetal lamb model [62], no causal genetic association between FGF signalling and HLHS was reported.
Transcription Factors			
HAND1	Ventricle development. LV specification. Ventricular trabeculation [63,64].	Defects in the left ventricle and endocardial cushions [65]. Defects in dorso-ventral patterning and interventricular septum formation [66].	A126fs frameshift mutation is identified in HLHS patient cardiac tissue [67], but this mutation does not cause HLHS in mice [68].
NKX2-5	Differentiation of CMs and ventricle formation. Purkinje fibre network, AV node, and bundle branch development [69].	Impaired looping morphogenesis [70] and atrial septal dysmorphogenesis [71].	Moderate: Cohort studies show a genetic variant (T178M) in a subset of HLHS [72]. Three different NKX2-5 mutations were identified in patients with ASD and VSD [73].
MYRF	Ventricular formation. Transcriptional regulation [74].	Hypoplastic ventricle [74].	Moderate: Associated with syndromic presentations of HLHS [75].
GATA4	Septation of chambers. CM differentiation. Valve formation [76].	GATA4 mutation (G295S) leads to thin ventricular myocardium and CM proliferation [77], and GATA4 has significant synergism with TBX5 required for early cardiogenesis [78].	Moderate: A heterozygous G296S missense mutation of GATA4 caused AVSDs and pulmonary valve stenosis in humans [79]. Additional novel point mutations in 3′-untranslated region of GATA4 gene were reported to be associated with sporadic non-syndromic AVSDs [80].
TBX1 [28]	Transcriptional control in cardiac progenitors in SHF [81].	Severe hypoplasia of SHF-dependent segments of the heart [81]. Aortic arch patterning defects and OFT defects in individuals with DiGeorge syndrome [82].	Weak: A 9 bp deletion DAGG379-381 was found to segregate with VSD [83].
TBX5	Cardiac septation. Conduction system. Chamber specification.	Holt–Oram syndrome (HOS) [84,85]. Dual knockdown of Tbx5 and Mef2c causes severe defects in heart tube looping [86].	Moderate: An intragenic duplication of TBX5 was reported in HOS patients presented with HLHS, AVSD, valve disease, and pulmonary stenosis [84].
WT1	Epicardial-to-mesenchymal transition [87].	Wilms tumour. Denys–Drash syndrome.	No causal genetic association between FGF signalling and HLHS was reported.
MEF2C	MEF2^+^ cardiac progenitor cells contribute to the endocardium and myocardium of the right ventricle, as well as the aortic and mitral valves during early cardiogenesis [88].	Loss of Mef2c function in the anterior second heart field results in a spectrum of outflow tract alignment defects [89].	A novel heterozygous missense mutation (pL38P) in MEF2C was identified in patients with PDA and VSD [90].
ISL1	SHF formation and regulation. Coronary development. CM lineage commitment [91].	OFT septation abnormalities, ASD, and VSD [91].	Moderate: Variations can increase susceptibility to CHD including HLHS [92]. A novel heterozygous missense mutation (pE137X) in ISL1 was identified in patients with PDA and VSD [93].
Structural/Regulatory Proteins			
MYH6	Encodes Alpha-myosin heavy chain. Predominantly controls atrial contractile function.	ASD. Late-onset hypertrophic cardiomyopathy.	Moderate: Multiple rare genetic variants (R443) found in HLHS cohorts [94,95]. MYH6 variant carriers exhibit impaired RA contractility [96]. HiPSC-CMs carrying an MYH6-R443P head domain variant display reduced contractility [97].
LRP2	Encodes endocytic receptors responsible for developmental signalling (SHH pathway) [98].	Neural tube defects. Donnai–Barrow syndrome [99].	Moderate: Multiple rare genetic variants found in HLHS genomic and transcriptomic studies [54].
GJA1 (Connexin43)	Encodes gap junctions. Facilitates electrical and metabolic communications between CMs.	Arrhythmogenic cardiomyopathy. Cardiac conduction disorders. Oculodentodigital dysplasia [100].	Emerging: One or more mutations were found in children with congenital heart malformation including HLHS [101]. Altered expression of GJA1 in HLHS heart tissue [102].

ASD, atrial septal defects; AV, atrioventricular; AVSD, atrioventricular septal defect; CM, cardiomyocytes; FHF, first heart field; LV, left ventricle; OFT, outflow tract; PDA, patent ductus arteriosus; SHF, second heart field; VSD, ventricular septal defects.

**Table 2 cells-14-01396-t002:** Clinical trials of stem cell applications in HLHS patients.

Study Name (NCT Number)	Stem Cell Type	Study Timeline	Study Stage	Enrolment: Total (Control/Treatment)	Route and Timing of Administration	Key Findings	Limitations	Status/Reference(s)
Transcoronary Infusion of Cardiac Progenitor Cells in Patients With Single Ventricle Physiology (TICAP) (NCT01273857)	Autologous CDC	2011–2013	Phase I	14 (7/7)	IC at Stage II/III surgical palliation	↑ RV function over 18 month period. ↑ HF status. Safety and feasibility of CDCs. Long-standing benefits on follow-up analysis.	Non-randomised; open-label; small sample size; variable timing of intervention.	Completed; Ishigami et al. [151].
Cardiac Progenitor Cell Infusion to Treat Univentricular Heart Disease (PERSEUS) (NCT01829750)	Autologous CDC	2013–2016	Phase II	34 (17/17)	IC at Stage II/III surgical palliation	↑ RV function; ↑ quality of life and somatic growth. Safety and feasibility of CDCs; long-standing benefits on follow-up analysis.	Limited long-term conclusion; open-label; single-ventricle; disease heterogeneity.	Completed; Ishigami et al. [152].
Safety Study of Autologous Umbilical Cord Blood Cells for Treatment of Hypoplastic Left Heart Syndrome (NCT01883076)	Autologous UCB-MNCs	2013–2021	Phase I	Phase I: 10 (0/10)	IM during Glenn operation (Stage II)	Preserved RV. Safety and feasibility.	Single-centre study; no control group; short-term follow-up; heterogeneity of UCB-MNCs.	Completed; Burkhart et al. [153].
Safety of Autologous Cord Blood Cells in HLHS Patients During Norwood Heart Surgery (NCT03431480)	Autologous UCB-MNCs	2018–2022	Phase I	10 (0/10)	IC during Norwood procedure	Preserved RV. Safety and feasibility.	Open-label; no control group; heterogeneity of UCB-MNCs.	Completed; Brizard et al. [154].
Intramyocardial Injection of Autologous Umbilical Cord Blood Derived Mononuclear Cells During Surgical Repair of Hypoplastic Left Heart Syndrome (NCT03779711)	Autologous UCB-MNCs	2019–2026	Phase IIb	95 (45/50)	IM at Stage II surgical repair	An unfavourable change in longitudinal cardiac strain and a greater incidence (20%) of at least one severe adverse event in treatment group. Failed to enhance cardiac functions.	Multicentre, open-label, non-randomised study; negative results.	Completed; Gallego-Navarro et al. [155].
Cardiac Stem/Progenitor Cell Infusion in Univentricular Physiology (APOLLON) (NCT02781922)	Autologous cardiac stem cells	2016–2023	Phase III	40 (NR/NR)	IC after Stage II/III surgical palliation	NR.	NR	NR.
Lomecel-B Injection in Patients With Hypoplastic Left Heart Syndrome: A Phase I/II Study (ELPIS) (NCT03525418/NCT04925024)	Allogeneic MSCs (Lomecel-B)	2018–2025	Phase I Phase II	Phase I: 10 (0/10) Phase II: 20 (10/10)	IM during Glenn operation	Phase I: Safety and feasibility. No alloimmune sensitisation. Phase II: NR.	Phase I: Open-label; no control; small sample size; non-randomised.	Phase I: Completed; Kaushal et al. [20]. Phase II: Ongoing.
Autologous Cardiac Stem Cell Injection in Patients With Hypoplastic Left Heart Syndrome: An Open Label Pilot Study (CHILD Trial) (NCT03406884)	Autologous c-kit+	2019–2024	Phase I/II	10 (Phase I); 22 (Phase II)	IM during Glenn operation	NR	Open-label.	Completed; pending report.
Mesoblast Stem Cell Therapy for Patients With Single Ventricle and Borderline Left Ventricle (NCT03079401)	Allogeneic MPCs	2017–2024	Phase I Phase II	19 (9/10)	IM during Glenn operation	NR	NR	Completed; pending report.

‘↑’, improved or increased; CDCs, cardiosphere-derived cells; IC, intracoronary; IM, intramyocardial; MPCs, mesenchymal precursor cells; MSCs, mesenchymal stem cells; NR, not reported; RV, right ventricle; UCB-MNCs, cord blood mononuclear cells.

**Table 3 cells-14-01396-t003:** HLHS-specific PSC-derived cardiomyocytes and functional characterisations.

Cell Source for iPSC Reprogramming	Reprogramming System	Genetic Mutations	Methodology (Medium, Small Molecular) for CM Differentiation	Key Findings	Applications	Reference
H7/H9 ESCs; WTC iPSCs; 176/1 iPSC 176/5 iPSC 176/8 iPSC	WiCell Research Institute; WiCell Research Institute; IMBA Stem Cell Core Facility	HAND1 knockout or NKX2.5 knockout PSCs	Day 0–2: CDM plus 30 ng/mL FGF2, 5 µM LY294002, 50 ng/mL AA, 10 ng/mL BMP4, and 1–4 µM CHIR99021 (Wnt-activation). Day 3–6: CDM plus BMP4 (10 ng/mL), FGF2 (8 ng/mL), insulin (10 µg/mL), 5 µM IWP2 (Wnt inhibition), and 0.5 µM RA. Day 7–8: CDM plus BMP4 (10 ng/mL), FGF2 (8 ng/mL), and insulin (10 µg/mL). Day 9 onwards: CDM plus insulin (10 µg/mL) for CM maintenance.	Differentiated CMs can be used to generate hollow, beating 3D structures, which was applied to confirm stage-specific regulation of HAND1 and NKX2.5 during cardiac development in the context of HLHS.	Cardioids generation, molecular insights into cardiac cavity morphogenesis, disease modelling such as cryoinjury (mimicking myocardial infarct) and HLHS	Hofbauer et al. [196]
Cardiac progenitor cells	Retroviruses to deliver OKSM factors	NR (not reported)	Matrigel-based monolayer: Day 0–1: RPMI/B27 medium supplemented with 100 ng/mL AA. Day 2–5: RPMI/B27 medium plus 10 ng/mL BMP 4.	HLHS iPSC-CMs exhibit a lower cardiomyogenic differentiation potential; decreased NKX2.5, TBX2, NOTCH/HEY signalling, and HAND1/2; and reduced H3K4 dimethylation and histone H3 acetylation but increased H3K27 trimethylation.	Provide molecular insights into complex transcriptional and epigenetic mechanisms underlying HLHS	Kobayashi et al. [141]
Dermal fibroblasts	Polycistronic lentiviral system for OCT4, KLF4, SOX2, and MYC (OKSM) factors	NR	EB-based protocol: Day 0–7: StemPro-34 SFM supplemented with 1 mM ascorbic acid, 4 × 10^−4^ M MTG, 10 ng/mL BMP4, 12.5 ng/mL bFGF, 6 ng/mL AA, 150 ng/mL DKK1, 5 ng/mL VEGF, and 5.4 μM SB-431542. Day 8 onwards: EBs were plated on 0.1% gelatin-coated 12-well culture plates (20 EBs per well) and cultured in StemPro-34 SFM supplemented with 10 ng/mL VEGF and 5 ng/mL bFGF.	HLHS iPSC-CMs display a decreased number of beating clusters; myofibrillar disorganisation, persistence of a foetal gene expression pattern, and changes in commitment to ventricular versus atrial lineages; different calcium transient patterns and electrophysiological responses.	NR	Jiang et al. [197]
Peripheral blood mononuclear cells (PBMCs)	CytoTune-iPSC 2.0 Sendai Reprogramming Kit	NOTCH1 knockout	Matrigel-based monolayer (over 90% confluency): Day 0–2: RPMI/B27 medium supplemented with 6 μM CHIR99021. Day 3: RPMI/B27 medium only. Day 4–5: RPMI/B27 medium plus 5 μM IWR-1. Day 6–9: Differentiated cells were incubated with RPMI1640 without glucose plus B27 supplement for 4 days to remove non-CMs. Day 10 onwards: RPMI/B27 until use.	HLHS iPSC-CMs confirm that disruption of NOTCH1 blocks human ventricular-like CM differentiation but promotes atrial-like CM generation, defective CM proliferation; impaired cell cycle progression and mitosis; and biased differentiation toward epicardial and SHF progenitors at the expense of FHF progenitors.	Possibly modelling HLHS, gaining new insights into the comprehension of the mechanisms underlying HLHS aetiology	Ye et al. [198]
Fibroblast (ATCC)	Episomal plasmid (ND2.0, NIH CRM control iPSC line)	Isogenic hiPSC hypomorphic NOTCH1 clones	Matrigel-based monolayer (over 90% confluency): CM differentiation media A and B.	HLHS iPSC-CMs display abnormalities in pathways associated with mitochondrial function, actin cytoskeleton, and cardiomyocyte development; skewed differentiation away from CMs and towards fibroblasts and SMCs; impaired cardiac cytoskeletal and mitochondrial architecture; and decreased CM contractility and ATP production.	Possibly modelling HLHS; high-throughput drug screening to identify potential HLHS drug such as auranofin	Lewis et al. [199]
Neonatal fibroblasts	Polycistronic vector encoding KLF4, OCT4, SOX2, as well as vectors encoding hc-Myc and hKlf4	Deleterious genetic variants of NOTCH1-4	Embryoid body (EB) based protocol: StemPro-34 SFM (basal media). Day 0–3: 10 ng/mL BMP4, 6 ng/mL AA, 5 ng/mL bFGF, 10 µM Y276321. Day 3–5: 150 ng/mL DKK1, 10 ng/mL VEGF, 5.4 µM SB431542, 0.25 µM Droso, 10 µM Y276321. Day 5–7: 150 ng/mL DKK1, 10 ng/mL VEGF, 10 µM Y276321. Day 7 onwards: EBs were plated on Matrigel-coated 12-well culture plates (20 EBs per well) and cultured in basal media plus 10 ng/mL VEGF and 5 ng/mL bFGF.	HLHS iPSC-CMs exhibit a reduced ability to give rise to mesodermal, cardiac progenitors, and mature CMs and an enhanced ability to differentiate to SMCs; lower beating rate; disorganised sarcomeres and sarcoplasmic reticulum; and blunted response to isoprenaline.	Possibly modelling HLHS; provide novel signalling and genetic insights into HLHS pathogenesis	Yang et al. [200]
Skin fibroblasts or PBMCs	CytoTune-iPSTM-iPS 2.0 Sendai Reprogramming Kit	heterozygous de-novo mutations in multiple genes including MYRF, BAI2, FGFR1, AIM1L, SYBU, MACF1, etc.	Matrigel-based monolayer (over 90% confluency): Day 1: CDM3 (RPMI1640 supplemented with 500 μM/mL Oryza sativa-derived recombinant human albumin and 213 μg/mL L-ascorbic acid 2-phosphate) plus 4–6 μM CHIR99021. Day 2–3: CDM3 plus 2 μM Wnt-C59. Day 4 onwards: CDM3 only.	Unique aberrations in autophagy terms were present when directing HLHS iPSCs toward early cardiac progenitors (CPs), whereas apoptosis-associated pathways appeared solely affected in later CPs (day 6) and cardiomyocytes (day 8); dysregulated lineage-specific CM differentiation; disrupted both early CM subtype lineage specification and CM differentiation and maturation in HLHS.	Possibly modelling HLHS; provide novel signalling and genetic insights into HLHS pathogenesis; use for 3D cardiac patch generation	Krane et al. [201]
Dermal fibroblasts	Lentiviral transduction of OKSM factors	Heterozygous NOTCH1(P1256L/P1964L)	Monolayer (over 90% confluency): Day 1: RPMI/B27 plus 40–100 ng/mL AA. Day 2–5: RPMI/B27 plus 5–20 ng/mL bFGF and BMP4. Day 5 onwards: RPMI/B27 only.	HLHS iPSC-CMS exhibit deficiency in NOTCH signalling pathway and a diminished capacity to generate CMs, as well as impaired NO signalling.	HLHS modelling; identification of small therapeutic molecules to compensate dysregulated NO signalling	Hrstka et al. [202] Theis et al. [203]
Dermal fibroblasts	Sendai Reprogramming Kits	MYH6-R443P variant	Geltrex-based monolayer: Day 0: mTeSR1 medium plus 5 μM ROCK inhibitor. Day 1–2: insulin-free RPMI/B27 plus 10 μM CHIR99021 and 10 ng/mL AA. Day 3–6: insulin-free RPMI/B27 with 5 μM IWP. Day 7 onwards: RPMI/B27 with insulin.	HLHS iPSC-CMs carrying the MYH6-R443P variant express beta-myosin heavy chain expression (MYH7), with impaired contractility, relaxation, and CM differentiation, as well as sarcomere disorganisation.	HLHS modelling; study how genetic variants contribute to HLHS	Kim et al. [97]; Tomita-Mitchell et al. [94]
NR	NR	NR	Small-molecule modulation of the canonical Wnt/β-catenin signalling pathway.	HLHS iPSC-CMs display impaired contractility; upregulation in sarcomere and cytoskeletal genes and downregulation in genes involved in mitochondrial function and metabolism; and reduced mitochondrial content, mitochondrial respiration, and oxidative metabolism.	HLHS modelling and drug screening	Paige et al. [204]
Fibroblasts or lymphoblastoid cells	Episomal plasmids encoding OSKM	Pathogenic variants associated with mitochondrial metabolism	Matrigel-based monolayer (over 90% confluency): Day 1–2: basal CDM3 media (RPMI 1640, BSA, B27, 213 μg/mL ascorbic acid) and 6 μM CHIR99021. Day 3–14: basal CDM3 media and 10 μM XAV939. Day 15 onwards: basal CDM3 media minus ascorbic acid.	HLHS iPSC-CMs exhibit impaired CM differentiation and contractile dysfunction; cell-cycle disturbance with metaphase arrest; increased CM apoptosis; myofibrillar disarray; mitochondrial dysfunction and perturbation of mitochondrial dynamics; and defects in YAP-regulated antioxidant response.	Modelling of HLHS and early heart failure; drug screening; identification of potential therapeutics such as sildenafil and TUDCA	Xu et al. [59]

AA, activin A; CDM, cardiac differentiation medium; EB, embryoid body; FHF, first heart field; MTG, monothioglycerol; SFM, serum free medium; SHF, second heart field; SMCs, smooth muscle cells.

## Data Availability

No new data were created or analyzed in this study.

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
