# Peer review of "Updated Applications of Stem Cells in Hypoplastic Left Heart Syndrome"

_cells, 2025, doi:10.3390/cells14171396_

Round 1
Reviewer 1 Report
Comments and Suggestions for Authors
In the manuscript entitled "Updated Applications of Stem Cells in Hypoplastic Left Heart Syndrome," the authors explored the evolving landscape of stem cell-based technologies, ranging from adult stem cells and induced pluripotent stem cells (iPSCs) to advanced 3D bioprinting and organoid models, as pivotal tools for elucidating HLHS pathophysiology and developing innovative therapeutic strategies.
This comprehensive review stands out for its rigorous and forward-looking analysis of stem cell applications in Hypoplastic Left Heart Syndrome (HLHS). The authors effectively demonstrate a deep understanding of the complex challenges posed by HLHS and present cutting-edge solutions offered by regenerative medicine.
The manuscript can be enhanced with the integration of cutting-edge research: this is a significant value to add. The recent finding that iPSCs constitutively express the progesterone receptor (PR) at the protein level (DOI: 10.1007/s12015-024-10776-6) is a crucial piece of information. This is a detail (refer to section 4.1: PSC-derived cardiomyocyte from HLHS patients) that adds a level of complexity but also opportunity to HLHS research, besides fidelity of iPSC-based models for physiological and pathological processes.
Author Response
Remarks to the Author:
In the manuscript entitled "Updated Applications of Stem Cells in Hypoplastic Left Heart Syndrome," the authors explored the evolving landscape of stem cell-based technologies, ranging from adult stem cells and induced pluripotent stem cells (iPSCs) to advanced 3D bioprinting and organoid models, as pivotal tools for elucidating HLHS pathophysiology and developing innovative therapeutic strategies.
This comprehensive review stands out for its rigorous and forward-looking analysis of stem cell applications in Hypoplastic Left Heart Syndrome (HLHS). The authors effectively demonstrate a deep understanding of the complex challenges posed by HLHS and present cutting-edge solutions offered by regenerative medicine.
Response: Thank you very much for your excellent comments about our work.
The manuscript can be enhanced with the integration of cutting-edge research: this is a significant value to add. The recent finding that iPSCs constitutively express the progesterone receptor (PR) at the protein level (DOI: 10.1007/s12015-024-10776-6) is a crucial piece of information. This is a detail (refer to section 4.1: PSC-derived cardiomyocyte from HLHS patients) that adds a level of complexity but also opportunity to HLHS research, besides fidelity of iPSC-based models for physiological and pathological processes.
Response: Thanks for your suggestion! We have critically analyzed this paper (DOI: 10.1007/s12015-024-10776-6) and found that although this study is very interesting, it has limited relevance with the scope of this review article. As per Journal (Cells) reference policy and also suggested by the editors at Cells, we respectively disagreed with you to include this reference into the revised manuscript.
Reviewer 2 Report
Comments and Suggestions for Authors
Updated Applications of Stem Cells in Hypoplastic Left Heart Syndrome
This is an interesting review that covers an important topic where there is considerable research interest. It is broad reaching covering the use of SC for therapeutic use, investigation of disease pathology and drug development. The review covers a lot of ground, however, there is not enough about HLHS itself (e.g. the heterogeneity of HLHS is mentioned but is not described – this is critical for thinking about what might cause it). There is very little about HLHS from a developmental point of view. What are thought to be the primary defects and how good is the evidence? Discussion of whether defects in the development myocardium, endocardium or valves are primary for HLHS (or all 3 in different cases) is crucial. This is very important when considering how the use of SC might help understanding the pathology. It is also important to discuss how almost all HLHS tissue from patients comes from stages where HLHS is already present, making it very difficult to know what happened early in the development of the malformation(s). There is little or no mention of animal models, although they have the potential to give information about how HLHS develops. I would strongly encourage more critical review of the referenced literature rather than taking it at face value. It is the complexity of HLHS that makes it so interesting and discussing this should not be avoided.
Specific comments:
Most of abstract is not about the review itself – it is background. It should be modified to better reflect the content of the review.
Line 41-63 – not convinced that this level of detail about surgical intervention is relevant to broader goals of this review. Similarly, Figure 1 does not seem relevant.
Line 101-107 - “These regions then give rise to the first heart field (FHF) and the second heart field (SHF). The FHF contributes to the formation of the LV and atria whilst the SHF contributes to the development of the RV and outflow tract. These two structures then subsequently fuse to form the cardiac crescent and the primitive heart tube. The primitive heart tube undergoes looping which transforms the structure from that of a linear tube into a structure with distinct chambers. Subsequent migration of the neural crest cells contributes to the formation of the aorticopulmonary septum [30] whilst endocardial cushions give rise to the valves and septa [31].” This is inaccurate in several respects. Neither heart fields nor chambers fuse in the cardiac crescent. NCC migrate into the cushions to form the aorticopulmonary septum – the two structures are intimately linked.
Line 110 – what developmental processes are disrupted to result in HLHS? This is critical information for the review, if it is known, and should be described. In fact, I don’t think it is known and this itself should be discussed. HLHS is clearly not a homogeneous malformation – there are likely different types, and thus there may be different causes for the small LV and associated malformations in different patients. This should be discussed.
Table 1 – “Animal models with NOTCH 1 defects display HLHS associated ventricular hypoplasia and valve abnormalities [37,38]”. These references do not support what is described in the table.
Moderate–BMP2/4 knockouts in mouse models lead to ventricular hypoplasia [42]; reference 42 is a review – the primary reference should be given in HLHS is observed.
Moderate–dysregulated signaling of pathways in HLHS iPSC-CMs. No reference given – should be provided.
A126fs frameshift mutation seen in patient cardiac tissue [60]. This does not cause HLHS when recapitulated in mice and therefore is questionable (Cardiovasc Res 2017 Dec 1;113(14):1732-1742. doi: 10.1093/cvr/cvx166.) This should be discussed here as well as later in the review.
Table 1 needs to be checked carefully. I do not want to go through all of these genes but the evidence is questionable in almost every case – I am not aware that many are functionally proven as causal for HLHS – this should be acknowledged. This lack of clear causality makes HLHS more interesting, not less.
Lines 120-140 – none of the data reported here provides any information of a role for Wnt signalling in HLHS. HLHS has left-sided lesions, whilst the SHF generally forms right-sided structures. How could a deficit in the latter lead to HLHS? I am not saying this is not possible, but if there is “direct implications in HLHS” then this should be described. Do embryos where Wnt signalling is disrupted have phenotypes that resemble HLHS for example? I am not aware that this is the case.
Lines for 140-150 – same as above for Wnt signalling, this paragraph provides no evidence for a role for RA signalling in HLHS pathogenesis. Do embryos where RA is supplemented or reduced have phenotypes that resemble HLHS?
Line 163 – I am not aware of any animal models with disruption of the Notch pathway having HLHS. Have patient’s variants been functionally verified? This should be discussed.
I don’t find this section very useful as no evidence of a direct link to HLHS is provided other than largely unvalidated patient variants.
Line 232 – “stem cell research has… expanded our treatment options for a variety of CHDs including HLHS over the past few decades”. This should be expanded upon. Specifically, how have treatment options been expanded?
Table 2 – the columns need to be broadened (or font reduced) so that the text can be read properly.
Section 3 on the use of SC in therapy is generally balanced and gives a good overview.
Line 401 – why do the authors suggest that differentiation of iPSC to CM is the desired outcome? What is the evidence that the primary defect in HLHS is CM? The widely supported “no flow, no grow” hypothesis (supported by the only anatomically accurate mouse model of HLHS – Rahman et al, 2021) would suggest that an abnormality in the developing valves may be the primary insult. This is a critical issue with all iPSC studies in HLHS so far – i.e. are they are in any way modelling what happens in the patient as HLHS develops? This must be discussed. As is, differentiation to CM is defended but other possibilities are not discussed.
Table 3 – I don’t think that this table works in the same way that Table 2 doesn’t – it is very hard to read the information in this format.
Line 425 –“… iPSCs and subsequently differentiating them into CMs [178,179], researchers are able to more accurately model HLHS in-vitro.” What is the evidence this is the case? How do we know that cardiac differentiation is disrupted in the early stages of HLHS development in patients? I don’t think we do know this. This does not say that the iPSC studies don’t have value, but caution should be taken when interpreting the results.
Line 440 – “In addition, HLHS iPSC-CMs 440 showed reduced contractility and altered calcium handling signaling, suggesting an intrinsic CM defect as a key part of HLHS pathogenesis rather than solely as a consequence of abnormal blood flow during fetal development.” This is crucial information. Reference 61 does not give any information about the phenotype of the HLHS patients and it is very difficult to tell if this data comes from one or several patients. These studies should be critically evaluated rather than taking the results as written in the abstract.
The published papers referred to in this section commonly show abnormalities in CM differentiation, contraction and down-regulation of CM markers – all factors that would be expected to affect the earliest stages of cardiac development. How does this fit with the observation that HLHS often develops over the second trimester, from an apparently initially normal heart (at 12-13 weeks of gestation)? Again, critical appraisal of the studies is required.
Line 462 – “the “no flow no grow” model is referred to here and is alluded to elsewhere. This should be described and the implications discussed.
Line 473 – is there potential to carry out iPSC studies following differentiation to endocardial or valve cells? This should be discussed.
Line 491 – the problem with the arguments made here is that HLHS is rarely detected before 20 of weeks of gestation, when the condition is well developed. The iPSC interventions described here are carried out on early cardiomyocytes – is there any equivalence between dish and patient in terms of treatment? What are the similarities and differences?
Lines 504-523 – is this relevant to HLHS?
Line 538 – can the authors be more specific about the maturity of iPSC-derived CM – how do they compare in fetal weeks?
Line 561 – discussion of the complexity of HLHS should be introduced earlier in the review as this has implications for all the points raised and discussed.
Lines 614-636 - Reference 203 refers to pericardial closure not a cardiac patch. It would be helpful to describe how patches – infused with SC or not - could be useful in the context of HLHS. Is the LV in HLHS in any equivalent to infarcted or heart failure myocardium in this context?
Line 654 – how might patches help with HLHS treatment – this needs to be explained.
Line 668 – “The emergence of cardiac organoids (COs) or cardioids has proven to be a better tool to model CHDs including HLHS.” What is the evidence for this? References should be given.
Line 673 – “Interestingly, some COs display properties of cardiac looping, polarity and even chamber formation similar to that of human embryonic heart development [215].” This statement greatly exceeds what the authors describe (or claim themselves) in this paper.
Line 684 – the role of Hand1 in HLHS is very controversial making the outcomes of this study (reference 169).
Figure 3B is too complex – is very difficult to interpret. I am not sure it is helpful in the context of the comment above.
Section 6.23 – explain how use of 3D printing could be useful specifically for HLHS studies – pros/cons compared to other technologies.
Author Response
Remarks to the Author:
This is an interesting review that covers an important topic where there is considerable research interest. It is broad reaching covering the use of SC for therapeutic use, investigation of disease pathology and drug development. The review covers a lot of ground, however, there is not enough about HLHS itself (e.g. the heterogeneity of HLHS is mentioned but is not described – this is critical for thinking about what might cause it). There is very little about HLHS from a developmental point of view. What are thought to be the primary defects and how good is the evidence? Discussion of whether defects in the development myocardium, endocardium or valves are primary for HLHS (or all 3 in different cases) is crucial. This is very important when considering how the use of SC might help understanding the pathology. It is also important to discuss how almost all HLHS tissue from patients comes from stages where HLHS is already present, making it very difficult to know what happened early in the development of the malformation(s). There is little or no mention of animal models, although they have the potential to give information about how HLHS develops. I would strongly encourage more critical review of the referenced literature rather than taking it at face value. It is the complexity of HLHS that makes it so interesting and discussing this should not be avoided.
Response: We are grateful for all constructive recommendations and positive comments that have helped us to hugely improve our manuscript. As suggested, we have now discussed all these aspects (the heterogeneity of HLHS; potential primary developmental defects of HLHS, the possible limitation with HLHS taken from patients, animal models of HLHS, etc) in the revised manuscript. Please see our point-by-point response detailed below and included in the revised manuscript. For your convenience, all the major changes were yellow-highlighted within the revised manuscript.
Specific comments:
Most of abstract is not about the review itself – it is background. It should be modified to better reflect the content of the review.
Response: As advised, we have now modified the abstract.
Line 41-63 – not convinced that this level of detail about surgical intervention is relevant to broader goals of this review. Similarly, Figure 1 does not seem relevant.
Response: We are grateful for your thoughtful suggestion. In this paragraph, we aimed to provide a brief summary of the current standard of care for HLHS patients, which in our opinion is important and sufficient to provide a simple foundation for the readers to have a good understanding about the most effective treatments that are currently available for HLHS patients in clinics. Together with the Figure 1, it provides a nice and brief overview regarding the traditional stage-specific surgical intervention and new development of stem cell application in HLHS as illustrated in Figure 1.
Line 101-107 - “These regions then give rise to the first heart field (FHF) and the second heart field (SHF). The FHF contributes to the formation of the LV and atria whilst the SHF contributes to the development of the RV and outflow tract. These two structures then subsequently fuse to form the cardiac crescent and the primitive heart tube. The primitive heart tube undergoes looping which transforms the structure from that of a linear tube into a structure with distinct chambers. Subsequent migration of the neural crest cells contributes to the formation of the aorticopulmonary septum [30] whilst endocardial cushions give rise to the valves and septa [31].” This is inaccurate in several respects. Neither heart fields nor chambers fuse in the cardiac crescent. NCC migrate into the cushions to form the aorticopulmonary septum – the two structures are intimately linked.
Response: Thanks a lot for your careful reading and pointing out the potential inaccuracy about some information. We have now re-written this paragraph (Page 4, 2nd paragraph).
Line 110 – what developmental processes are disrupted to result in HLHS? This is critical information for the review, if it is known, and should be described. In fact, I don’t think it is known and this itself should be discussed. HLHS is clearly not a homogeneous malformation – there are likely different types, and thus there may be different causes for the small LV and associated malformations in different patients. This should be discussed.
Response: Thank you very much for this important suggestion. We have now included such discussion in the revises manuscript (Page 4, 3rd paragraph; Page 12, 3rd paragraph).
Table 1 – “Animal models with NOTCH 1 defects display HLHS associated ventricular hypoplasia and valve abnormalities [37,38]”. These references do not support what is described in the table.
Moderate–BMP2/4 knockouts in mouse models lead to ventricular hypoplasia [42]; reference 42 is a review – the primary reference should be given in HLHS is observed.
Moderate–dysregulated signaling of pathways in HLHS iPSC-CMs. No reference given – should be provided.
A126fs frameshift mutation seen in patient cardiac tissue [60]. This does not cause HLHS when recapitulated in mice and therefore is questionable (Cardiovasc Res 2017 Dec 1;113(14):1732-1742. doi: 10.1093/cvr/cvx166.) This should be discussed here as well as later in the review.
Table 1 needs to be checked carefully. I do not want to go through all of these genes but the evidence is questionable in almost every case – I am not aware that many are functionally proven as causal for HLHS – this should be acknowledged. This lack of clear causality makes HLHS more interesting, not less.
Response: We have carefully checked all the information presented in this table to ensure their accuracy. We also corrected relevant content and cited correct/appropriate references to support the key findings included in Table 1 (modified Table 1).
Lines 120-140 – none of the data reported here provides any information of a role for Wnt signalling in HLHS. HLHS has left-sided lesions, whilst the SHF generally forms right-sided structures. How could a deficit in the latter lead to HLHS? I am not saying this is not possible, but if there is “direct implications in HLHS” then this should be described. Do embryos where Wnt signalling is disrupted have phenotypes that resemble HLHS for example? I am not aware that this is the case.
Response: As suggested, we have now carefully revised this section (Page 9, 2nd paragraph).
Lines for 140-150 – same as above for Wnt signalling, this paragraph provides no evidence for a role for RA signalling in HLHS pathogenesis. Do embryos where RA is supplemented or reduced have phenotypes that resemble HLHS?
Response: Thank you very much for your constructive suggestion. We have now carefully revised this section (Page 9-10, last and first paragraph).
Line 163 – I am not aware of any animal models with disruption of the Notch pathway having HLHS. Have patient’s variants been functionally verified? This should be discussed.
Response: Thanks for your thoughtful suggestion. We have now discussed the potential issues with Notch signalling (Page 10, 2nd paragraph).
I don’t find this section very useful as no evidence of a direct link to HLHS is provided other than largely unvalidated patient variants.
Response: We assumed that you were talking about the section regarding BMP signalling. We do believe that disruption of BMP signaling could potentially cause one or more cardiac developmental defects observed in HLHS. We have now provided more information about the potential role of BMP signalling in cardiac development by discussing some genetic studies related to BMP signalling and cardiac developmental defects (Page 10, last paragraph).
Line 232 – “stem cell research has… expanded our treatment options for a variety of CHDs including HLHS over the past few decades”. This should be expanded upon. Specifically, how have treatment options been expanded?
Response: To make clear, we have now modified this statement as ‘Indeed, stem cell research has significantly advanced our understanding of human disease etiology and a variety of stem cells have been used as an adjunctive therapy for some congenital disease including HLHS in the past decades as discussed in the following sections’ (Page 13, 1st paragraph).
Table 2 – the columns need to be broadened (or font reduced) so that the text can be read properly.
Response: As suggested, we have now modified it to increase its readability.
Section 3 on the use of SC in therapy is generally balanced and gives a good overview.
Response: Thanks for your positive comment about this section.
Line 401 – why do the authors suggest that differentiation of iPSC to CM is the desired outcome? What is the evidence that the primary defect in HLHS is CM? The widely supported “no flow, no grow” hypothesis (supported by the only anatomically accurate mouse model of HLHS – Rahman et al, 2021) would suggest that an abnormality in the developing valves may be the primary insult. This is a critical issue with all iPSC studies in HLHS so far – i.e. are they are in any way modelling what happens in the patient as HLHS develops? This must be discussed. As is, differentiation to CM is defended but other possibilities are not discussed.
Response: Thanks for your thoughtful suggestion. We have now discussed the potential limitation using iPSC-derived CMs as the main cellular model to study HLHS in the current literature (Page 27, 2nd paragraph). Additionally, the “no flow, no grow” hypothesis has also been discussed in the revised manuscript (Page 4, 3rd paragraph and page 27, 2nd paragraph).
Table 3 – I don’t think that this table works in the same way that Table 2 doesn’t – it is very hard to read the information in this format.
Response: As advised, we have now modified this table to improve its readability (modified Table 3).
Line 425 –“… iPSCs and subsequently differentiating them into CMs [178,179], researchers are able to more accurately model HLHS in-vitro.” What is the evidence this is the case? How do we know that cardiac differentiation is disrupted in the early stages of HLHS development in patients? I don’t think we do know this. This does not say that the iPSC studies don’t have value, but caution should be taken when interpreting the results.
Response: We have now rephrased this statement as ‘By reprogramming dermal fibroblasts or other cells from HLHS patients into iPSCs and subsequently differentiating them into CMs and other cardiac cells[205,206], researchers are able to model HLHS in-vitro’ (Page 24, 1st paragraph).
Line 440 – “In addition, HLHS iPSC-CMs showed reduced contractility and altered calcium handling signaling, suggesting an intrinsic CM defect as a key part of HLHS pathogenesis rather than solely as a consequence of abnormal blood flow during fetal development.” This is crucial information. Reference 61 does not give any information about the phenotype of the HLHS patients and it is very difficult to tell if this data comes from one or several patients. These studies should be critically evaluated rather than taking the results as written in the abstract.
Response: Thanks for your thoughtful suggestion. We have now modified our statement about this study (Page 24, end of 1st paragraph).
The published papers referred to in this section commonly show abnormalities in CM differentiation, contraction and down-regulation of CM markers – all factors that would be expected to affect the earliest stages of cardiac development. How does this fit with the observation that HLHS often develops over the second trimester, from an apparently initially normal heart (at 12-13 weeks of gestation)? Again, critical appraisal of the studies is required.
Response: Thanks a lot for raising this interesting point. Actually, it has somehow widely suggested that the primary defect of HLHS occurs at a much earlier time point (between weeks 4 and 8 of gestation), which could result in the abnormal development of the left side of the heart. While some HLHS-associated developmental defects may become more evident in the second trimester and routine ultrasounds can detect it, the fundamental structural changes begin much earlier due to issues like aortic or mitral valve problems, which prevent the left-sided heart structures from growing properly (PMID: 32119463; Bookshelf ID: NBK554576; https://www.hopkinsmedicine.org/health/conditions-and-diseases/hypoplastic-left-heart-syndrome-hlhs-in-children; https://www.childrenshospital.org/conditions/hypoplastic-left-heart-syndrome).
Line 462 – “the “no flow no grow” model is referred to here and is alluded to elsewhere. This should be described and the implications discussed.
Response: As suggested, we have now discussed the “no flow no grow” model in section-2 (Page 4, 3rd paragraph).
Line 473 – is there potential to carry out iPSC studies following differentiation to endocardial or valve cells? This should be discussed.
Response: We have now mentioned one study about iPSC-derived endocardial cells for studying HLHS and also discussed potential contributions of other iPSC-derived cardiac cells including valve cells in HLHS pathogenesis (Ref-216, page 27, 2nd paragraph).
Line 491 – the problem with the arguments made here is that HLHS is rarely detected before 20 of weeks of gestation, when the condition is well developed. The iPSC interventions described here are carried out on early cardiomyocytes – is there any equivalence between dish and patient in terms of treatment? What are the similarities and differences?
Response: As suggested, we have now discussed this issue as one of the potential limitations with iPSC-CMs studies (Page 27, end of 2nd paragraph). For the detection of HLHS, please refer to our previous response to a similar comment.
Lines 504-523 – is this relevant to HLHS?
Response: Yes, we do need to consider drug toxicity when developing any potential therapeutics for any disease including HLHS since drug toxicity is the major reason for drug withdraw from markets and iPSC is a useful model to early predict drug toxicity before approval for clinical use.
Line 538 – can the authors be more specific about the maturity of iPSC-derived CM – how do they compare in fetal weeks?
Response: Thanks for raising this interesting point. We have now added some information about the commonly reported immaturity of iPSC-CMS (Page 27, beginning of the 3rd paragraph). Although iPSC-CMs has been described to exhibit a fetal-like CM phenotype, unfortunately, no study has compared the iPSC-CMs with fetal developmental stages/weeks so far.
Line 561 – discussion of the complexity of HLHS should be introduced earlier in the review as this has implications for all the points raised and discussed.
Response: As advised, we have now discussed it in section 2.2. (Page 12, 3rd paragraph).
Lines 614-636 - Reference 203 refers to pericardial closure not a cardiac patch. It would be helpful to describe how patches – infused with SC or not - could be useful in the context of HLHS. Is the LV in HLHS in any equivalent to infarcted or heart failure myocardium in this context?
Response: Thanks for your careful reading. We have now deleted this reference. We also discussed the potential application of cardiac patches in HLHS (Page 30, end of 2nd and 3rd paragraph).
Line 654 – how might patches help with HLHS treatment – this needs to be explained.
Response: Please refer to our response to your last comment.
Line 668 – “The emergence of cardiac organoids (COs) or cardioids has proven to be a better tool to model CHDs including HLHS.” What is the evidence for this? References should be given.
Response: We have now rephrased this statement to ‘Recent developments have seen the emergence of cardiac organoids (COs) or cardioids which are used to advance our understanding of CHDs including HLHS.’ (Page 31, 3rd paragraph).
Line 673 – “Interestingly, some COs display properties of cardiac looping, polarity and even chamber formation similar to that of human embryonic heart development [215].” This statement greatly exceeds what the authors describe (or claim themselves) in this paper.
Response: We have now rephrased this statement to ‘Interestingly, some PSC-derived COs could recapitulate patterns of early cardiomyogenesis, resembling early embryonic heart anlagen and nascent fore-gut endoderm[243].’ (Page 31, 3rd paragraph).
Line 684 – the role of Hand1 in HLHS is very controversial making the outcomes of this study (reference 169).
Response: In our opinion, the key finding from this study discussed here is consistent with those observed in Hand1 transgenic mice (PMID: 15143159). In that study, Togi et al reported that Hand1/eHAND plays an important role in cardiac chamber formation, but it is not a master regulatory gene that specifies the left ventricle myocyte lineage.
Figure 3B is too complex – is very difficult to interpret. I am not sure it is helpful in the context of the comment above.
Response: We have now simplified Figure 3B to make it easy to understand (Modified Figure 3B).
Section 6.23 – explain how use of 3D printing could be useful specifically for HLHS studies – pros/cons compared to other technologies.
Response: As suggested, we have now discussed how 3D bioprinting can be used in HLHS (Page 36, end of 2nd paragraph). Moreover, the main purpose of this section is proposing a potential application of 3D in HLHS in the future, there is currently no reported studies in this aspect in literature. Therefore, it is impossible for us to compare and discuss any PROS/CONS between 3D bioprinting and other technologies in the context of HLHS.